# DOSE: Diffusion Dropout with Adaptive Prior for Speech Enhancement

**Wenxin Tai**[1], **Yue Lei**[1], **Fan Zhou**[1,2]*, **Goce Trajcevski**[3], **Ting Zhong**[1,2]
University of Electronic Science and Technology of China
Kashi Institute of Electronics and Information Industry
Iowa State University

## Abstract

Speech enhancement (SE) aims to improve the intelligibility and quality of speech in the presence of non-stationary additive noise. Deterministic deep learning models have traditionally been used for SE, but recent studies have shown that generative approaches, such as denoising diffusion probabilistic models (DDPMs), can also be effective. However, incorporating condition information into DDPMs for SE remains a challenge. We propose a *model-agnostic* method called DOSE that employs two efficient condition-augmentation techniques to address this challenge, based on two key insights: (1) We force the model to prioritize the condition factor when generating samples by training it with dropout operation; (2) We inject the condition information into the sampling process by providing an informative adaptive prior. Experiments demonstrate that our approach yields substantial improvements in high-quality and stable speech generation, consistency with the condition factor, and inference efficiency. Codes are publicly available at `https://github.com/ICDM-UESTC/DOSE`.

## 1 Introduction

Speech enhancement (SE) aims to improve the intelligibility and quality of speech, particularly in scenarios where degradation is caused by non-stationary additive noise. It has significant practical implications in various fields such as telecommunications [1], medicine [2], and entertainment [3]. Modern deep learning models are often used to learn a deterministic mapping from noisy to clean speech. While deterministic models have long been regarded as more powerful in the field of SE, recent advancements in generative models [4, 5] have significantly closed this gap.

One such generative approach is based on using denoising diffusion probabilistic models (DDPMs) [6, 7], which have been shown to effectively synthesize natural-sounding speech. Several diffusion enhancement models have been developed [4, 5, 8], which try to learn a probability distribution over the data and then generate clean speech conditioned on the noisy input. A key challenge in using diffusion enhancement models is how to effectively incorporate condition information into learning and generating faithful speech [9, 10, 8]. Previous works address this issue through designing specific condition-injecting strategies [4, 9, 11] or devising complex network architectures [10, 5].

We conduct a thorough examination to understand the limitation of diffusion-based SE methods and find that diffusion enhancement models are susceptible to *condition collapse*, where the primary cause of inconsistent generation is the *non-dominant position of the condition factor*. We thus introduce a new paradigm to effectively incorporate condition information into the diffusion enhancement models. Specifically, we propose a Diffusion-drOpout Speech Enhancement method (DOSE), which is a model-agnostic SE method (Figure 1) that employs two efficient condition-augmentation tech-

---

*Corresponding author: `fan.zhou@uestc.edu.cn`

37th Conference on Neural Information Processing Systems (NeurIPS 2023).

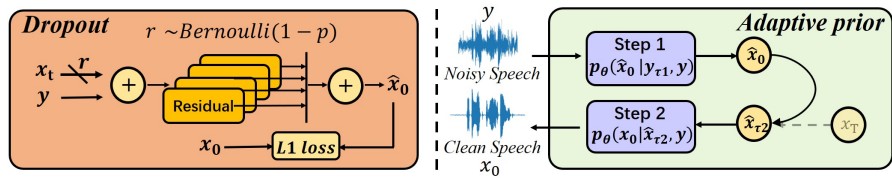

Figure 1: An illustration of the proposed DOSE. DOSE consists two primary procedures: (1) training a condition diffusion model using dropout operation, and (2) generating speech using a conditional diffusion model equipped with the adaptive prior.

niques: (1) During training, we randomly drop out intermediate-generated samples. This dropout mechanism guides the model's attention toward the condition factors; (2) Instead of letting the model generate samples from scratch (Gaussian distribution), we employ an adaptive prior derived from the conditional factor to generate samples. Experiments on benchmark datasets demonstrate that our method surpasses recent diffusion enhancement models in terms of both accuracy and efficiency. Additionally, DOSE produces more natural-sounding speech and exhibits stronger generalization capabilities compared to deterministic mapping-based methods using the same network architecture.

## 2 Related works

There are two main categories of diffusion-based SE methods: (1) designing specific condition-injecting strategies [4, 9, 11, 5], or (2) generating speech with an auxiliary condition optimizer[12, 10, 8]. The first category considerates noisy speech in the diffusion (or reverse) process, either by linearly interpolating between clean and noisy speech along the process [4, 9], or by defining such a transformation within the drift term of a stochastic differential equation (SDE) [11, 5].

Works from the second category rely on an auxiliary condition optimizer – a generator (diffusion model) synthesizes clean speech and a condition optimizer informs what to generate [12, 10, 8]. Both the generator and condition optimizer have the ability to denoise, with the latter undertaking the core part. Given the challenges in leveraging condition information [8], diffusion-based SE methods within this category often necessitate specific network architecture design to guarantee the participation of condition factors.

In a paradigm sense, our method is quite similar but different to the second branch – unlike previous approaches that require additional auxiliary networks, DOSE is an end-to-end diffusion-based SE method. In addition, DOSE is model-agnostic that does not need any specific network design to guarantee consistency between the generated sample and its corresponding condition factor.

## 3 Preliminaries

We now provide a brief introduction to the diffusion probabilistic model (diffusion models, for short), the definition of speech enhancement, and the condition collapse problem.

### 3.1 Diffusion models

A diffusion model [13, 6] consists of a forward (or, diffusion) process and a reverse process. Given a data point $\boldsymbol{x}_0$ with probability distribution $p(\boldsymbol{x}_0)$, the forward process gradually destroys its data structure by repeated application of the following Markov diffusion kernel:

$$p(\boldsymbol{x}_t|\boldsymbol{x}_{t-1}) = \mathcal{N}(\boldsymbol{x}_t; \sqrt{1-\beta_t}\boldsymbol{x}_{t-1}, \beta_t\boldsymbol{I}), \quad t \in \{1, 2, \cdots, T\}, \tag{1}$$

where $\beta_1, \cdots, \beta_T$ is a pre-defined noise variance schedule. With enough diffusion step $T$, $p(\boldsymbol{x}_T)$ converges to the unit spherical Gaussian distribution. Based on the Markov chain, the marginal distribution at arbitrary timestep $t$ has the following analytical form:

$$p(\boldsymbol{x}_t|\boldsymbol{x}_0) = \mathcal{N}(\boldsymbol{x}_t; \sqrt{\bar{\alpha}_t}\boldsymbol{x}_0, (1-\bar{\alpha}_t)\boldsymbol{I}), \quad t \in \{1, 2, \cdots, T\}, \tag{2}$$

where $\bar{\alpha}_t = \prod_{s=1}^{t}(1-\beta_s)$.

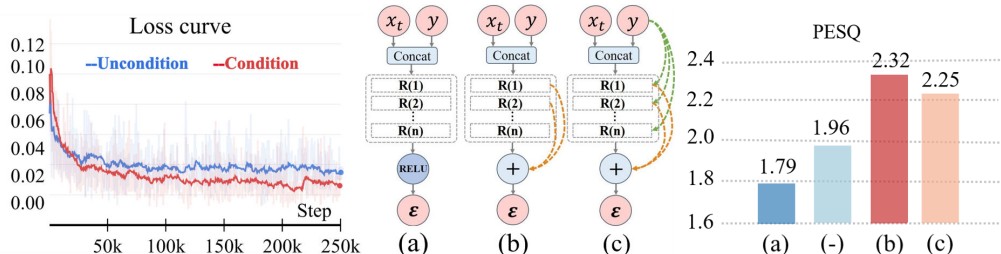

Figure 2: Investigation of the condition collapse problem. From left to right: (1) comparison of loss curves between unconditional and conditional diffusion models; (2) three variants; (3) PESQ performance of different variants, (-) represent the unprocessed speech.

As for the reverse process, it aims to learn a transition kernel from $\boldsymbol{x}_t$ to $\boldsymbol{x}_{t-1}$, which is defined as the following Gaussian distribution [6]:

$$p_{\boldsymbol{\theta}}(\boldsymbol{x}_{t-1}|\boldsymbol{x}_t) = \mathcal{N}(\boldsymbol{x}_{t-1}; \boldsymbol{\mu}_{\boldsymbol{\theta}}(\boldsymbol{x}_t, t), \boldsymbol{\Sigma}(\boldsymbol{x}_t, t)), \tag{3}$$

where $\boldsymbol{\theta}$ is the learnable parameter and $\boldsymbol{\mu}_{\boldsymbol{\theta}}(\boldsymbol{x}_t, t) = \frac{1}{\sqrt{1-\beta_t}}(\boldsymbol{x}_t - \frac{\beta_t}{\sqrt{1-\bar{\alpha}_t}}\boldsymbol{\epsilon}_{\boldsymbol{\theta}}(\boldsymbol{x}_t, t))$ denotes the mean of $\boldsymbol{x}_{t-1}$, which is obtained by subtracting the estimated Gaussian noise $\boldsymbol{\epsilon}_{\boldsymbol{\theta}}(\boldsymbol{x}_t, t)$ in the $\boldsymbol{x}_t$. With such a learned transition kernel, one can approximate the data distribution $p(\boldsymbol{x}_0)$ via:

$$p_{\boldsymbol{\theta}}(\boldsymbol{x}_0) = \int p_{\boldsymbol{\theta}}(\boldsymbol{x}_T) \prod_{t=1}^{T} p_{\boldsymbol{\theta}}(\boldsymbol{x}_{t-1}|\boldsymbol{x}_t) d\boldsymbol{x}_{1:T}, \tag{4}$$

where $p_{\boldsymbol{\theta}}(\boldsymbol{x}_T) = \mathcal{N}(\boldsymbol{x}_T; \boldsymbol{0}, \boldsymbol{I})$.

## 3.2 Problem formulation

Speech enhancement refers to methods that try to reduce distortions, make speech sounds more pleasant, and improve intelligibility. In real environments, the monaural noisy speech $\boldsymbol{y}$ in the time domain can be modeled as:

$$\boldsymbol{y} = \boldsymbol{x} + \boldsymbol{n} \tag{5}$$

where $\boldsymbol{x}$ and $\boldsymbol{n}$ denote clean and noise signals, respectively. For human perception, the primary goal of speech enhancement is to extract $\boldsymbol{x}$ from $\boldsymbol{y}$. Mapping-based speech enhancement methods directly optimize $p_{\boldsymbol{\theta}}(\boldsymbol{x}|\boldsymbol{y})$, while diffusion enhancement methods generate clean samples through a Markov process $p_{\boldsymbol{\theta}}(\boldsymbol{x}_{0:T-1}|\boldsymbol{x}_{1:T}, \boldsymbol{y})$.

## 3.3 Condition Collapse in diffusion enhancement models

The *condition collapse* problem in speech enhancement was first proposed in [8] and it refers to the limited involvement of the condition factor during conditional diffusion training, resulting in inconsistencies between the generated speech and its condition factor.

In this work, we argue that the condition factor $\boldsymbol{y}$ indeed participates and helps the intermediate-generated sample $\boldsymbol{x}_t$ approximate $p(\boldsymbol{x}_{t-1}|\boldsymbol{x}_t, \boldsymbol{x}_0)$. Our assertion is supported by the experiment depicted in the left part of Figure 2 – the diffusion model equipped with the condition factor exhibits a lower loss curve compared to the unconditional one[2]. To better understand the condition collapse phenomenon, we devise two variants that explicitly modify the mutual information between the condition factor and the model's output (Figure 2 (middle)). We use skip connections to add the condition factor to multiple layers, forcing the likelihood of maintaining a strong connection between the condition factor and output features. Since the dependence of the output on any hidden state in the hierarchy becomes weaker as one moves further away from the output in that hierarchy (cf. [14]), using skip connections can explicitly enhance connections between the generated sample and condition factor.

---

[2]We use DiffWave [7] as basic architecture and use the same experimental settings as [4, 9] – the only difference being the change in the way of condition-injecting since most speech enhancement methods will directly use noisy speech as the condition factor, rather than Mel-spectrogram.

As shown in Figure 2 (right), an increase in mutual information (connections) leads to a significant improvement in the consistency between the generated sample and the condition factor ($a \rightarrow b$). However, it requires a meticulously designed model to guarantee its effectiveness ($b \rightarrow c$). While previous studies [5, 10, 8] focus on explicitly enhancing the consistency between the output speech and condition factor through specific network architecture design, we explore the possibility of a solution independent of the model architecture. This would broaden the applicability of our method, as it enables slight modifications to existing deterministic mapping-based models to transform them into diffusion enhancement models.

## 4  Methodology

Considering the diffusion model provides a transition function from $\boldsymbol{x}_t$ to $\boldsymbol{x}_{t-1}$, typical condition generation process is represented as:

$$p_{\boldsymbol{\theta}}(\boldsymbol{x}_0|\boldsymbol{y}) = \int \underbrace{p(\boldsymbol{x}_T)}_{\text{Prior}} \prod_{t=1}^{T} \underbrace{p_{\boldsymbol{\theta}}(\boldsymbol{x}_{t-1}|\boldsymbol{x}_t, \boldsymbol{y})}_{\text{Condition}} d\boldsymbol{x}_{1:T}, \quad \boldsymbol{x}_T \sim \mathcal{N}(\boldsymbol{x}_T; \boldsymbol{0}, \boldsymbol{I}). \tag{6}$$

Our experiments above indicate that $p_{\boldsymbol{\theta}}(\boldsymbol{x}_{t-1}|\boldsymbol{x}_t, \boldsymbol{y})$ will easily collapse to $p_{\boldsymbol{\theta}}(\boldsymbol{x}_{t-1}|\boldsymbol{x}_t)$, resulting in the condition generation process degenerating into a vanilla unconditional process:

$$\int p_{\boldsymbol{\theta}}(\boldsymbol{x}_T) \prod_{t=1}^{T} p_{\boldsymbol{\theta}}(\boldsymbol{x}_{t-1}|\boldsymbol{x}_t, y) d\boldsymbol{x}_{1:T} \Rightarrow \int p_{\boldsymbol{\theta}}(\boldsymbol{x}_T) \prod_{t=1}^{T} p_{\boldsymbol{\theta}}(\boldsymbol{x}_{t-1}|\boldsymbol{x}_t) d\boldsymbol{x}_{1:T}. \tag{7}$$

As a result, facilitating automatic learning of the joint distribution for both clean and noisy speech samples does not work well for the speech enhancement task.

### 4.1  Condition augmentation I: Adaptive Prior

Let's revisit Eq. (6): since we cannot easily inject the condition factor into the condition term, how about the prior term? For example, we can modify the condition generation process as:

$$p_{\boldsymbol{\theta}}(\boldsymbol{x}_0|\boldsymbol{y}) = \int \underbrace{p(\boldsymbol{x}_\tau|\boldsymbol{y})}_{\text{Conditional}} \prod_{t=1}^{\tau} \underbrace{p_{\boldsymbol{\theta}}(\boldsymbol{x}_{t-1}|\boldsymbol{x}_t)}_{\text{Unconditional}} d\boldsymbol{x}_{1:\tau}, \tag{8}$$

where $p(\boldsymbol{x}_\tau|\boldsymbol{y})$ is formulated as $p(\boldsymbol{x}_\tau|\boldsymbol{y}) = \mathcal{N}(\boldsymbol{x}_\tau; \sqrt{\bar{\alpha}_\tau}\boldsymbol{y}, (1-\bar{\alpha}_\tau)\boldsymbol{I})$. The following propositions verify the feasibility of our proposal.

**Proposition 1.** *For any $\xi > 0$ such that $0 < \xi < M$ for some finite positive value $M$, there exists a positive value $\tau \in \{0, \cdots, T\}$ that satisfies:*

$$D_{KL}\left(p(\boldsymbol{x}_t|\boldsymbol{x}) \| p(\boldsymbol{x}_t|\boldsymbol{y})\right) \leq \xi, \quad \forall \tau \leq t \leq T, \tag{9}$$

*where $p(\boldsymbol{x}_t|\boldsymbol{c}) = \mathcal{N}(\boldsymbol{x}_t; \sqrt{\bar{\alpha}_t}\boldsymbol{c}, (1-\bar{\alpha}_t)\boldsymbol{I})$.*

**Remark 1.** *This proposition indicates that, given a tolerable margin of error $\xi$ and a well-trained diffusion model, we can always find a suitable $\tau$ such that we are able to recover the clean speech $\boldsymbol{x}$ from its noisy one $\boldsymbol{y}$ using Eq. (8).*

While **Proposition 1** allows us to generate clean speech $\boldsymbol{x}$ given the noisy speech $\boldsymbol{y}$ using Eq. (8), it does not guarantee that our model will achieve successful recovery with a high probability.

**Proposition 2.** *Let $\boldsymbol{x}$ be the clean sample, $\boldsymbol{y}$ be it's corresponding noisy one, and $\boldsymbol{x}'$ be any neighbor from the neighbor set $\mathcal{S}(\boldsymbol{x})$. Then diffusion enhancement models can recover $\boldsymbol{x}$ with a high probability if the following inequality is satisfied:*

$$\log\left(\frac{p(\boldsymbol{x})}{p(\boldsymbol{x}')}\right) > \frac{1}{2\sigma_t^2}\left(\|\boldsymbol{x}-\boldsymbol{y}\|_2^2 - \|\boldsymbol{x}'-\boldsymbol{y}\|_2^2\right), \quad \forall \boldsymbol{x}' \in \mathcal{S}(\boldsymbol{x}), \tag{10}$$

*where $\sigma_t^2 = \frac{1-\bar{\alpha}_t}{\bar{\alpha}_t}$.*

**Remark 2.** *Assuming that the condition factor $\boldsymbol{y}$ is closer to $\boldsymbol{x}$ than to $\boldsymbol{x}'$, we obtain a non-positive right-hand side (RHS). For a given $\boldsymbol{x}$, the left-hand side (LHS) value is fixed, and to ensure the inequality always holds, a smaller $\sigma_t^2$ is preferred.*

As shown in Figure 3, $\sigma_t^2$ will increase as the timestep $t$ increases. Thus, according to **Proposition 2**, we should choose a small $\tau$ for Eq. (8) to maximize the probability of successfully recovering the clean speech from the noisy one. However, constrained by **Proposition 1**, $\tau$ cannot be too small. In other words, the clean speech distribution $p(\boldsymbol{x}_\tau)$ and the noisy speech distribution $p(\boldsymbol{y}_\tau)$ will get closer over the forward diffusion process, and the gap $|\boldsymbol{n}| = |\boldsymbol{y} - \boldsymbol{x}|$ between the noisy speech and the clean one will indeed be "washed out" by the increasingly added noise. Since the original semantic information will also be removed if $\tau$ is too large, there should be a trade-off when we set $\tau$ for the diffusion enhancement model.

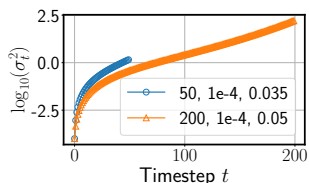

Figure 3: The change curves of $\log_{10} \sigma_t^2$. Elements in legend are $T, \beta_1, \beta_T$ respectively.

**Condition optimizer.** We find that both propositions are correlated with the condition factor $\boldsymbol{y}$. If we can reduce the gap between the condition factor and clean speech, we can choose a smaller $\tau$, effectively increasing the likelihood of recovering clean speech. One simple idea is to employ a neural network $f_{\boldsymbol{\psi}}$ to optimize the condition factor, as demonstrated in [15]. Accordingly, we can rewrite Eq. (8) as:

$$p_{\boldsymbol{\theta}, \boldsymbol{\psi}}(\boldsymbol{x}_0 | \boldsymbol{y}) = \int p_{\boldsymbol{\psi}}(\boldsymbol{x}_\tau | \boldsymbol{y}) \prod_{t=1}^{\tau} p_{\boldsymbol{\theta}}(\boldsymbol{x}_{t-1} | \boldsymbol{x}_t) d\boldsymbol{x}_{1:\tau}, \tag{11}$$

where $p_{\boldsymbol{\psi}}(\boldsymbol{x}_\tau | \boldsymbol{y}) = \mathcal{N}(\boldsymbol{x}_\tau; \sqrt{\bar{\alpha}_\tau} f_{\boldsymbol{\psi}}(\boldsymbol{y}), (1 - \bar{\alpha}_\tau) \boldsymbol{I})$.

In practice, we should also consider failure cases of the condition optimizer in complex scenarios, especially the issue of excessive suppression that has been reported in recent literature [16, 17, 18]. To mitigate this issue, we use $0.5\boldsymbol{c} + 0.5\boldsymbol{y}$ (like a simple residual layer) as a mild version of the condition factor:

$$p_{\boldsymbol{\psi}}(\boldsymbol{x}_\tau | \boldsymbol{y}) = \mathcal{N}(\boldsymbol{x}_\tau; 0.5\sqrt{\bar{\alpha}_\tau} (f_{\boldsymbol{\psi}}(\boldsymbol{y}) + \boldsymbol{y}), (1 - \bar{\alpha}_\tau) \boldsymbol{I}). \tag{12}$$

We call $p_{\boldsymbol{\psi}}(\boldsymbol{x}_\tau | \boldsymbol{y})$ the adaptive prior as it varies with different noisy samples $\boldsymbol{y}$.

### 4.2 Condition augmentation II: Diffusion Dropout

Aside from changing the prior $p(\boldsymbol{x}_T)$ to conditional prior $p_{\boldsymbol{\psi}}(\boldsymbol{x}_\tau | \boldsymbol{y})$, we also optimize the condition term $p_{\boldsymbol{\theta}}(\boldsymbol{x}_{t-1} | \boldsymbol{x}_t, \boldsymbol{y})$. Instead of designing specific condition-injecting strategies [4, 9, 11] or devising complicated network architecture [8, 10, 5], we attempt to "do subtraction" by discarding some shared (intermediate-generated samples & condition factor) and important (target-related) information from intermediate-generated samples. Naturally, if we discard some information from $\boldsymbol{x}_t$, then the diffusion enhancement model is forced to use the condition factor $\boldsymbol{y}$ to recover the speech. Taking a further step, we can even discard the entire $\boldsymbol{x}_t$, as the condition factor $\boldsymbol{y}$ alone is sufficient for recovering the clean speech $\boldsymbol{x}_0$ (this is what deterministic models do). To this end, we define a neural network $f_{\boldsymbol{\theta}}(d(\boldsymbol{x}_t, p), \boldsymbol{y}, t)$ to approximate $p(\boldsymbol{x}_{t-1} | \boldsymbol{x}_t, \boldsymbol{x}_0)$:

$$p_{\boldsymbol{\theta}}(\boldsymbol{x}_{t-1} | \boldsymbol{x}_t, \boldsymbol{y}) = \mathcal{N}(\boldsymbol{x}_{t-1}; f_{\boldsymbol{\theta}}(d(\boldsymbol{x}_t, p), \boldsymbol{y}, t), \boldsymbol{\Sigma}(\boldsymbol{x}_t, t)), \tag{13}$$

where $d(\boldsymbol{x}_t, p)$ is the dropout operation:

$$d(\boldsymbol{x}_t, p) = \begin{cases} \boldsymbol{x}_t & \text{if } r = 1 \\ \boldsymbol{\epsilon} & \text{if } r = 0 \end{cases}, \quad r \sim \text{Bernoulli}(1 - p). \tag{14}$$

### 4.3 DOSE training

Ho et al. [6] and much of the following work choose to parameterize the denoising model through directly predicting $\boldsymbol{\epsilon}$ with a neural network $\boldsymbol{\epsilon}_{\boldsymbol{\theta}}(\boldsymbol{x}_t, t)$, which implicitly sets:

$$\hat{\boldsymbol{x}}_0 = \frac{1}{\sqrt{\bar{\alpha}_t}} \left( \boldsymbol{x}_t - \sqrt{1 - \bar{\alpha}_t} \boldsymbol{\epsilon}_{\boldsymbol{\theta}} \right). \tag{15}$$

In this case, the training loss is also usually defined as the mean squared error in the $\boldsymbol{\epsilon}$-space $\|\boldsymbol{\epsilon} - \boldsymbol{\epsilon}_{\boldsymbol{\theta}}(\boldsymbol{x}_t, t)\|_2^2$. Although this standard specification works well for training an unconditional diffusion model, it is not suited for DOSE – for two reasons.

| **Algorithm 1** DOSE Training | **Algorithm 2** DOSE Sampling |
|---|---|
| 1: **choose** $p$ | 1: **choose** $\tau_1, \tau_2, \tau_2 < \tau_1 \le T$ |
| 2: **repeat** | 2: Step 1: Generate $\hat{\boldsymbol{x}}_{\tau_2}$ |
| 3:    $\boldsymbol{x}_0 \sim p(\boldsymbol{x})$ | 3:    $\boldsymbol{y}_{\tau_1} \sim \mathcal{N}(\sqrt{\bar{\alpha}_{\tau_1}}\boldsymbol{y}, (1-\bar{\alpha}_{\tau_1})\boldsymbol{I})$ |
| 4:    $t \sim \text{Uniform}(\{1,\ldots,T\})$ | 4:    $\hat{\boldsymbol{x}}_0 = f_{\boldsymbol{\theta}}\left(\boldsymbol{y}_{\tau_1}, \boldsymbol{y}, \tau_1\right)$ |
| 5:    $\epsilon \sim \mathcal{N}(0, I)$ | 5:    $\hat{\boldsymbol{x}}_{\tau_2} \sim \mathcal{N}(0.5\sqrt{\bar{\alpha}_{\tau_2}}(\hat{\boldsymbol{x}}_0 + \boldsymbol{y}), (1-\bar{\alpha}_{\tau_2})\boldsymbol{I})$ |
| 6:    Take gradient descent step on | 6: Step 2: Generate $\hat{\boldsymbol{x}}_0$ |
| 7:      $\nabla_{\boldsymbol{\theta}}\|\boldsymbol{x}_0 - \boldsymbol{x}_{\boldsymbol{\theta}}(d\left(\boldsymbol{x}_t, p\right), \boldsymbol{y}, t)\|_2^2$ | 7:    $\hat{\boldsymbol{x}}_0 = f_{\boldsymbol{\theta}}\left(\hat{\boldsymbol{x}}_{\tau_2}, \boldsymbol{y}, \tau_2\right)$ |
| 8: **until converged** | 8: **return** $\hat{\boldsymbol{x}}_0$ |

First, we cannot estimate $\epsilon$ without the help of $\boldsymbol{x}_t$ because $\epsilon$ and $\boldsymbol{y}$ are independent. Second, as discussed earlier, we want DOSE to start with a small timestep and we strive to make $\tau$ small. However, as $\tau$ approaches zero, small changes in x-space have an increasingly amplified effect on the implied prediction in $\epsilon$-space (Eq. (15)). In other words, the efforts made by diffusion enhancement models become so negligible that diffusion models lose their ability to calibrate the speech at small timesteps.

So, we need to ensure that the estimation of $\hat{\boldsymbol{x}}_0$ remains flexible as the timestep $t$ gets smaller. Considering the equivalently of the $\epsilon$-space loss $\|\epsilon - \epsilon_{\boldsymbol{\theta}}(\boldsymbol{x}_t, t)\|_2^2$ to a weighted reconstruction loss in x-space $\frac{1}{\sigma_t^2}\|\boldsymbol{x}_0 - \boldsymbol{x}_{\boldsymbol{\theta}}(\boldsymbol{x}_t, t)\|_2^2$, we can directly estimate the clean speech $\boldsymbol{x}_0$ at each timestep $t$:

$$\mathcal{L} = \mathbb{E}_{\boldsymbol{x}_0 \sim p(\boldsymbol{x}), t \in \{1,\ldots,T\}} \left[\|\boldsymbol{x}_0 - f_{\boldsymbol{\theta}}(d(\boldsymbol{x}_t, p), \boldsymbol{y}, t)\|_2^2\right] \tag{16}$$

### 4.4 DOSE inference

After training, the ideal scenario is that $p_{\boldsymbol{\theta}}(\boldsymbol{x}_{t-1}|\boldsymbol{x}_t, \boldsymbol{y})$ approximates $p(\boldsymbol{x}_{t-1}|\boldsymbol{x}_t, \boldsymbol{x}_0)$ precisely, enabling us to generate clean speech using Eq. (6). However, when applied in practice, it is difficult to completely eliminate errors (both sample error and true error). If these errors are not effectively managed or corrected during the generation process, the quality of the generated samples may deteriorate, leading to artifacts, blurriness, etc [19, 20]. This issue is particularly pronounced when using diffusion models for fine-grained conditional generation tasks, as diffusion models require a large number of steps to generate samples, which will significantly reduce the consistency between the generated sample and its condition factor (see §5.3, Figure 6).

The adaptive prior (Sec 4.1) provides an opportunity to address the error accumulation issue. Specifically, we can select a suitable $\tau$ smaller than $T$, conditioned on an adaptive prior, and generate speech in fewer steps. We can extend Eq. 11 by transforming the unconditional diffusion enhancement model into a conditional one:

$$p_{\boldsymbol{\theta}, \boldsymbol{\psi}}(\boldsymbol{x}_0|y) = \int p_{\boldsymbol{\psi}}(\boldsymbol{x}_\tau|\boldsymbol{y}) \prod_{t=1}^{\tau} p_{\boldsymbol{\theta}}(\boldsymbol{x}_{t-1}|\boldsymbol{x}_t, \boldsymbol{y}) d\boldsymbol{x}_{1:\tau}, \tag{17}$$

and the number of sampling steps is reduced from $T$ to $\tau + 1$.

Readers familiar with diffusion models may recall that the standard process repeatedly applies a "single-step" denoising operation $\boldsymbol{x}_{t-1} = denoise(\boldsymbol{x}_t; t)$ that aims to convert a noisy sample at some timestep $t$ to a (slightly less) noisy sample at the previous timestep $t - 1$. In fact, each application of the one-step denoiser consists of two steps: (1) an estimation of the fully denoised sample $\boldsymbol{x}_0$ from the current timestep $t$, and (2) computing a (properly weighted, according to the diffusion model) average between this estimated denoised sample and the noisy sample at the previous timestep $t - 1$. Thus, instead of performing the entire $\tau$-step diffusion process to denoise a sample, it is also possible to run $denoise$ once and simply output the estimated sample in one shot [21]. Accordingly, Eq. (17) can be further rewritten as:

$$p_{\boldsymbol{\theta}, \boldsymbol{\psi}}(\boldsymbol{x}_0|\boldsymbol{y}) = \int p_{\boldsymbol{\psi}}(\boldsymbol{x}_\tau|\boldsymbol{y}) p_{\boldsymbol{\theta}}(\boldsymbol{x}_0|\boldsymbol{x}_\tau, \boldsymbol{y}) d\boldsymbol{x}_\tau \tag{18}$$

We can even achieve DOSE without the condition optimizer $f_{\boldsymbol{\psi}}(\cdot)$ – using conditional diffusion enhancement model instead. For example, we can generate clean speech via:

$$p_{\boldsymbol{\theta}}(\boldsymbol{x}_0|\boldsymbol{y}) = \int \int p_{\boldsymbol{\theta}}(\hat{\boldsymbol{x}}_{\tau_2}|\boldsymbol{y}_{\tau_1}, \boldsymbol{y}) p_{\boldsymbol{\theta}}(\boldsymbol{x}_0|\hat{\boldsymbol{x}}_{\tau_2}, \boldsymbol{y}) d\hat{\boldsymbol{x}}_{\tau_2} d\boldsymbol{y}_{\tau_1}, \tag{19}$$

where $\tau_1, \tau_2$ $(\tau_2 < \tau_1 \leq T)$ are two pre-defined hyper-parameters. The motivation behind Eq. (19) is that, once we have trained a neural network $f_{\boldsymbol{\theta}}(\boldsymbol{x}_t, \boldsymbol{y}, t)$ that can accurately estimate $\boldsymbol{x}_0$ (Eq. (16)), according to the theoretical analysis in Sec 4.1, we can first choose a suitable value for $\tau_1$ to ensure a relatively good approximation of $\boldsymbol{x}_0$:

$$\hat{\boldsymbol{x}}_0 = f_{\boldsymbol{\theta}}(\boldsymbol{y}_{\tau_1}, \boldsymbol{y}, \tau_1) \approx f_{\boldsymbol{\theta}}(\boldsymbol{x}_{\tau_1}, \boldsymbol{y}, \tau_1) \tag{20}$$

In the second step, once we have obtained a good condition factor, we can choose a smaller timestep $\tau_2 < \tau_1$ to get a better estimation of $\boldsymbol{x}_0$ than $\hat{\boldsymbol{x}}_0$ generated in the first step.

**Summary.** DOSE has three important benefits: (1) By dropping $\boldsymbol{x}_t$ entirely, we make the condition factor $\boldsymbol{y}$ the "protagonist", automatically enhancing the consistency between the generated sample and the condition factor. (2) By training the model with this modified training objective, DOSE can perform well not only on Gaussian noise ($\boldsymbol{x}_t \to \boldsymbol{x}_0$) but also on various types of non-Gaussian noise ($\boldsymbol{y} \to \boldsymbol{x}_0$). (3) DOSE is efficient (2 steps), faster than existing diffusion enhancement models.

## 5 Experiments

We compare DOSE with prevailing diffusion enhancement methods and deterministic mapping-based enhancement methods in §5.1. We conduct a counterfactual verification to understand the intrinsic mechanism of DOSE in §5.2. We show two visual cases of excessive suppression and error accumulation in §5.3. While providing a self-contained version of our main results, we note that we also have additional quantitative observations reported in the Appendices. Specifically, we compare DOSE with other baselines via subjective evaluation (Appendix A.2); We investigate the significance of the proposed adaptive prior and explain why we need to use a mild version of the condition factor (Appendix A.3); We examine the effect of our new training objective and demonstrate the necessity of using it (Appendix A.4); We explain why we use two steps in speech generation (Appendix A.5); We provide parameter sensitivity experiments (Appendix A.6); We show plenty of visual cases of excessive suppression and error accumulation (Appendix A.7 and A.8). To help readers better understand our research, we include a discussion subsection in Appendix A.9. Specifically, we: (1) Analyze the reasons behind the superior generalizability of diffusion enhancement models compared to deterministic mapping-based models (from the robust training perspective); (2) Explain why we use 0.5 in the mild version of the condition factor; (3) Discuss the broader impacts of speech enhancement methods.

**Dataset and baselines.** Following previous works [4, 9, 8], we use the VoiceBank-DEMAND dataset [22, 23] for performance evaluations. To investigate the generalization ability of models, we use CHiME-4 [24] as another test dataset following [9], i.e., the models are trained on VoiceBank-DEMAND and evaluated on CHiME-4. We compare our model with recent open-sourced diffusion enhancement models such as DiffuSE [4], CDiffuSE [9], SGMSE [11], SGMSE+ [5], and DR-DiffuSE [8]. Since the only difference between SGMSE+ and SGMSE is their network architecture, we compare our model with just one of them.

**Evaluation metrics.** We use the following metrics to evaluate SE performance: the perceptual evaluation of speech quality (PESQ) [25], short-time objective intelligibility (STOI) [26], segmental signal-to-noise ratio (SSNR), the mean opinion score (MOS) prediction of the speech signal distortion (CSIG) [27], the MOS prediction of the intrusiveness of background noise (CBAK) [27] and the MOS prediction of the overall effect (COVL) [27]. Besides these metrics, we also design two MOS metrics (MOS and Similarity MOS) for subjective evaluation.

**Configurations.** To ensure a fair comparison, we keep the model architecture exactly the same as that of the DiffWave model [7] for all methods[3]. DiffWave takes 50 steps with the linearly spaced training noise schedule $\beta_t \in [1 \times 10^{-4}, 0.035]$ [4]. We train all methods for 300,000 iterations using 1 NVIDIA RTX 3090 GPU with a batch size of 16 audios. We select the best values for $\tau_1$ and $\tau_2$ according to the performance on a validation dataset, a small subset (10%) extracted from the training data. More experiment settings can be found in Appendix A.10.

---

[3]Since the focus of our work is on studying the capabilities of diffusion dropout and adaptive prior for consistency enhancement, we use off-the-shelf architectures to avoid confounding our findings with model improvements. This decision (using DiffWave) rests on both its widely validated effectiveness and the minimal changes it required in the baseline experimental setup.

Table 1: Comparison of different diffusion enhancement methods.

| Method | Year | Efficiency | Dataset | STOI(%)↑ | PESQ↑ | CSIG↑ | CBAK↑ | COVL↑ |
|---|---|---|---|---|---|---|---|---|
| Unprocessed | – | – | | 92.1 | 1.97 | 3.35 | 2.44 | 2.63 |
| DiffWave | 2021 | 1 step (dis) | | 93.3 | 2.51 | 3.72 | 3.27 | 3.11 |
| DiffuSE | 2021 | 6 steps | | 93.5$^{+0.20}_{\pm0.05}$ | 2.39$^{-0.12}_{\pm0.01}$ | 3.71$^{-0.01}_{\pm0.01}$ | 3.04$^{-0.23}_{\pm0.01}$ | 3.03$^{-0.08}_{\pm0.01}$ |
| CDiffuSE | 2022 | 6 steps | VB | 93.7$^{+0.40}_{\pm0.05}$ | 2.43$^{-0.08}_{\pm0.01}$ | 3.77$^{+0.05}_{\pm0.01}$ | 3.09$^{-0.18}_{\pm0.01}$ | 3.09$^{-0.02}_{\pm0.01}$ |
| SGMSE | 2022 | 50 steps | | 93.3$^{+0.00}_{\pm0.08}$ | 2.34$^{-0.17}_{\pm0.01}$ | 3.69$^{-0.03}_{\pm0.01}$ | 2.90$^{-0.37}_{\pm0.01}$ | 3.00$^{-0.11}_{\pm0.01}$ |
| DR-DiffuSE | 2023 | 6 steps | | 92.9$^{-0.04}_{\pm0.06}$ | 2.50$^{-0.01}_{\pm0.02}$ | 3.68$^{-0.04}_{\pm0.02}$ | 3.27$^{+0.00}_{\pm0.02}$ | 3.08$^{-0.03}_{\pm0.02}$ |
| DOSE | – | 2 steps | | 93.6$^{+0.30}_{\pm0.05}$ | 2.56$^{+0.05}_{\pm0.01}$ | 3.83$^{+0.11}_{\pm0.01}$ | 3.27$^{+0.00}_{\pm0.01}$ | 3.19$^{+0.08}_{\pm0.01}$ |
| Unprocessed | – | – | | 71.5 | 1.21 | 2.18 | 1.97 | 1.62 |
| DiffWave | 2021 | 1 step (dis) | | 72.3 | 1.22 | 2.21 | 1.95 | 1.63 |
| DiffuSE | 2021 | 6 steps | | 83.7$^{+11.4}_{\pm0.05}$ | 1.59$^{+0.36}_{\pm0.01}$ | 2.91$^{+0.70}_{\pm0.01}$ | 2.19$^{+0.24}_{\pm0.01}$ | 2.19$^{+0.56}_{\pm0.01}$ |
| CDiffuSE | 2022 | 6 steps | CHIME-4 | 82.8$^{+10.5}_{\pm0.05}$ | 1.58$^{+0.36}_{\pm0.01}$ | 2.88$^{+0.67}_{\pm0.01}$ | 2.15$^{+0.20}_{\pm0.01}$ | 2.18$^{+0.55}_{\pm0.01}$ |
| SGMSE | 2022 | 50 steps | | 84.5$^{+12.2}_{\pm0.05}$ | 1.57$^{+0.34}_{\pm0.02}$ | 2.92$^{+0.71}_{\pm0.01}$ | 2.18$^{+0.23}_{\pm0.02}$ | 2.18$^{+0.55}_{\pm0.01}$ |
| DR-DiffuSE | 2023 | 6 steps | | 77.6$^{+5.30}_{\pm0.06}$ | 1.29$^{+0.07}_{\pm0.04}$ | 2.40$^{+0.19}_{\pm0.02}$ | 2.04$^{+0.09}_{\pm0.01}$ | 1.78$^{+0.15}_{\pm0.01}$ |
| DOSE | – | 2 steps | | 86.6$^{+14.3}_{\pm0.05}$ | 1.52$^{+0.30}_{\pm0.01}$ | 2.71$^{+0.50}_{\pm0.01}$ | 2.15$^{+0.20}_{\pm0.01}$ | 2.06$^{+0.43}_{\pm0.01}$ |

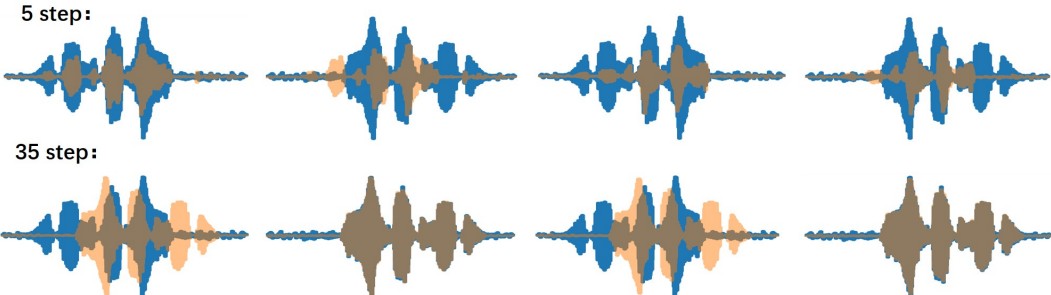

5 step：

35 step：

Figure 4: Counterfactual visualization. The first two columns are associated with a dropout probability of 0.1, while the last two columns are associated with a dropout probability of 0.9. In each row, the blue waveforms in the first and third columns are the counterfactual samples, and the blue waveforms in the second and fourth columns are the normal samples. The orange waveforms are generated samples from the model.

## 5.1 Performance comparison

We compare our method with previous diffusion enhancement methods and summarize our experimental results in Table 1. We observe that: **(1)** Diffusion enhancement methods have better generalizability than deterministic methods. **(2)** Methods with specific condition-injecting strategies, such as DiffuSE, CDiffuSE, and SGMSE, have strong generalization but perform slightly worse than deterministic mapping-based methods in matched scenarios. **(3)** Method (DR-DiffuSE) with auxiliary condition optimizer, performs better in matched scenarios and shows a slight improvement in mismatched scenarios. **(4)** Our method performs well in both matched and mismatched scenarios and is on par with state-of-the-art diffusion enhancement models while requiring fewer steps.

## 5.2 Counterfactual verification

We perform a counterfactual verification to gain insights into the underlying mechanism of DOSE. To verify whether dropout can increase the "discourse power" of the conditional factor, we keep the condition factor $y$ fixed and reverse the intermediate-generated speech at a specific step ($reverse(x_t)$). This reversed intermediate-generated speech is called a counterfactual sample. Notably, if the final generated speech is more similar to the condition factor than the counterfactual speech, we can conclude that the condition factor plays a dominant role in the generation process. Otherwise, we can say that the condition factor is less influential.

As shown in Figure 4, we compare the performance of two models with different dropout probabilities (0.1 vs. 0.9). We have two findings here: **(1)** A higher dropout probability encourages the model to prioritize the condition factor even with a small timestep $t$. **(2)** When timestep $t$ is large, DOSE ef-

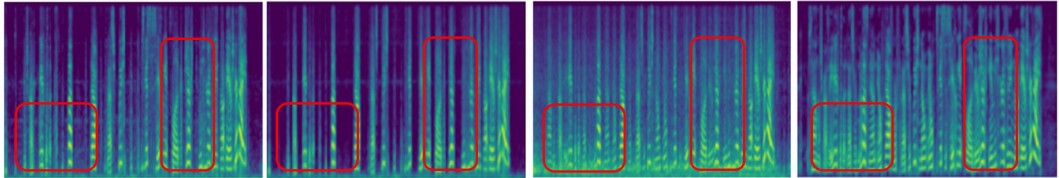

Figure 5: Excessive suppression visualization (unconditional diffusion enhancement model on CHIME-4). From left to right: (1) DiffWave (dis); (2) adaptive prior with the estimated condition; (3) adaptive prior with the mild condition; (4) clean speech.

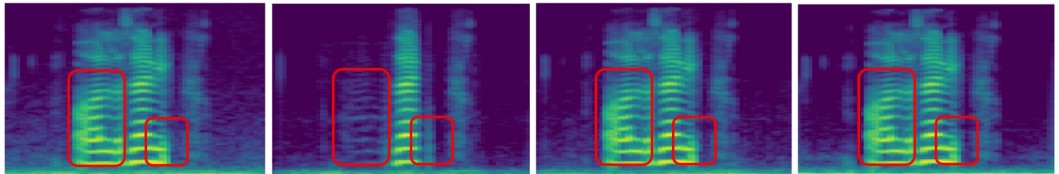

Figure 6: Error accumulation visualization (VB, DOSE). From left to right: (1) noisy speech; (2) full (50) steps; (3) 2 steps; (4) clean.

fectively captures condition information, ensuring the model's robustness to noise and maintaining consistency in the early stages of inference.

### 5.3 Excessive suppression & error accumulation

We provide a visual case of excessive suppression in Figure 5 and a visual case of error accumulation in Figure 6. From Figure 5, we can see that: **(1)** The deterministic model fails in mismatched scenarios and generates samples that lose speech details; **(2)** The diffusion enhancement model generate defective speech when directly using the estimated speech as the condition factor; **(3)** The diffusion enhancement model equipped with a mild version of the condition factor can recover clean speech effectively. From Figure 6, we notice that: **(1)** Full-step generation can remove noise and generate natural-sounding speech. However, it can't guarantee the consistency between the generated speech and condition factor; **(2)** Two-step speech generation with adaptive prior can promise consistency and high quality simultaneously.

## 6  Conclusions

In this work, we present a new approach DOSE that effectively incorporates condition information into diffusion models for speech enhancement. DOSE uses two efficient condition-augmentation techniques to address the condition collapse problem. Comprehensive experiments on benchmark datasets demonstrate the efficiency and effectiveness of our method.

In our method, there are two groups of hyper-parameters: the dropout probability $p$ for the dropout operation and two timesteps $\tau_1, \tau_2$ for the adaptive prior. These parameters are critical to model performance. For example, if the dropout probability is set too high, the diffusion enhancement model will rely solely on the condition factor to estimate the speech. Then our diffusion enhancement model will degenerate into a deterministic model, losing its generalizability. We also need to make a trade-off when choosing the timestep $\tau$ (especially $\tau_1$): On one hand, a large $\tau$ is needed to reduce the gap between the clean speech and condition factor. On the other hand, the original semantic information will also be removed if $\tau$ is set too large.

In practice, it is necessary to evaluate the model on a subset of data and then empirically set the hyperparameters. These manually defined hyper-parameters are selected based on the Empirical Risk Minimization (ERM) principle and may not be optimal for every individual sample. Thus, an important direction for future research is to develop methods that can adaptively choose hyper-parameters for different samples. It is also expected that the model can adaptively select appropriate coefficients when forming a mild version of the conditioning factor.

## Acknowledgement

This work was supported in part by National Natural Science Foundation of China (Grant No.62176043 and No.62072077), Natural Science Foundation of Sichuan Province (Grant No.2022NSFSC0505), Kashgar Science and Technology Bureau (Grant No.KS2023025), and National Science Foundation SWIFT (Grant No.2030249).

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

# A Appendix

In the supplemental material,

- **A.1:** We provide the proofs for Propositions 1 and 2.

- **A.2:** We compare DOSE with other baselines via subjective evaluation.

- **A.3:** We investigate the significance of the proposed adaptive prior and explain why we need to use a mild version of the condition factor.

- **A.4:** We examine the effect of our new training objective and demonstrate the necessity of using it.

- **A.5:** We explain why we use two steps in speech generation.

- **A.6:** We provide parameter sensitivity experiments.

- **A.7:** We present several visual cases of excessive suppression.

- **A.8:** We present several visual cases of error accumulation.

- **A.9:** We analyze the reasons behind the superior generalizability of diffusion enhancement models compared to deterministic mapping-based models (from the robust training perspective), explain why we use 0.5 in the mild version of the condition factor, and discuss the broader impacts of speech enhancement methods.

- **A.10:** We include more information about speech processing and basic architecture.

## A.1 Mathematical proofs

We now present the proofs of the Propositions stated in the main text.

**Proposition 1.** (cf. §4.1): *For any $\xi > 0$ such that $0 < \xi < M$ for some finite positive value $M$, there exists a positive value $\tau \in \{0, \cdots, T\}$ that satisfies:*

$$D_{KL}\left(p(\boldsymbol{x}_t|\boldsymbol{x})\|p(\boldsymbol{x}_t|\boldsymbol{y})\right) \leq \xi, \quad \forall \tau \leq t \leq T, \tag{9}$$

*where $p(\boldsymbol{x}_t|\boldsymbol{c}) = \mathcal{N}(\boldsymbol{x}_t; \sqrt{\bar{\alpha}_t}\boldsymbol{c}, (1 - \bar{\alpha}_t)\boldsymbol{I})$.*

*Proof.* Given two Gaussian distributions $P, Q$ defined over a vector space $\mathbb{R}^d$, the KL divergence of multivariate Gaussian distributions is defined as follows:

$$D_{KL}\left(P\|Q\right) = \frac{1}{2}\left(\text{Tr}\left(\boldsymbol{\Sigma}_2^{-1}\boldsymbol{\Sigma}_1\right) + (\boldsymbol{\mu}_2 - \boldsymbol{\mu}_1)^T\boldsymbol{\Sigma}_2^{-1}(\boldsymbol{\mu}_2 - \boldsymbol{\mu}_1) - d + \ln\left(\frac{\det\boldsymbol{\Sigma}_2}{\det\boldsymbol{\Sigma}_1}\right)\right). \tag{21}$$

Here, $\boldsymbol{\mu}_1 \in \mathbb{R}^d$ and $\boldsymbol{\Sigma}_1 \in \mathbb{R}^{d \times d}$ are the mean and covariance matrix of distribution $P$, and $\boldsymbol{\mu}_2 \in \mathbb{R}^d$ and $\boldsymbol{\Sigma}_2 \in \mathbb{R}^{d \times d}$ are the mean and covariance matrix of distribution $Q$. $d$ is the dimensionality of the vectors (i.e., the number of dimensions in the vector space), and Tr denotes the trace operator.

Note that when the two Gaussian distributions have diagonal covariance matrices (i.e., when the different dimensions are independent), the above formula simplifies to the sum of the KL divergences of each univariate Gaussian distribution. Thus, given two Gaussian distributions $p(\boldsymbol{x}_t|\boldsymbol{x})$, $p(\boldsymbol{x}_t|\boldsymbol{y})$ and Eq. (5), the KL divergence between these two distributions can be calculated as follows:

$$D_{KL}\left(p(\boldsymbol{x}_t|\boldsymbol{x})\|p(\boldsymbol{x}_t|\boldsymbol{y})\right) = \frac{\bar{\alpha}_t}{1 - \bar{\alpha}_t}\|\boldsymbol{y} - \boldsymbol{x}\|_2^2 = \frac{1}{\sigma_t^2}\|\boldsymbol{n}\|_2^2. \tag{22}$$

According to the definition of the diffusion model and Figure 3, $D_{KL}\left(p(\boldsymbol{x}_t|\boldsymbol{x})\|p(\boldsymbol{x}_t|\boldsymbol{y})\right)$ is a monotonically decreasing function. Ideally, for a bounded error $\boldsymbol{n}$ with (almost) infinite timestep $T$, we have:

$$\lim_{t \to 0} D_{KL}\left(p(\boldsymbol{x}_t|\boldsymbol{x})\|p(\boldsymbol{x}_t|\boldsymbol{y})\right) = +\infty, \quad \lim_{t \to T} D_{KL}\left(p(\boldsymbol{x}_t|\boldsymbol{x})\|p(\boldsymbol{x}_t|\boldsymbol{y})\right) = 0. \tag{23}$$

According to Bolzano's theorem, there exists at least one point $\tau$ in the interval $\{0, \cdots, T\}$ such that $D_{KL}\left(p(\boldsymbol{x}_\tau|\boldsymbol{x})\|p(\boldsymbol{x}_\tau|\boldsymbol{y})\right) = \xi$. Then, Eq. (9) holds for $\tau \leq t \leq T$. $\qquad \square$

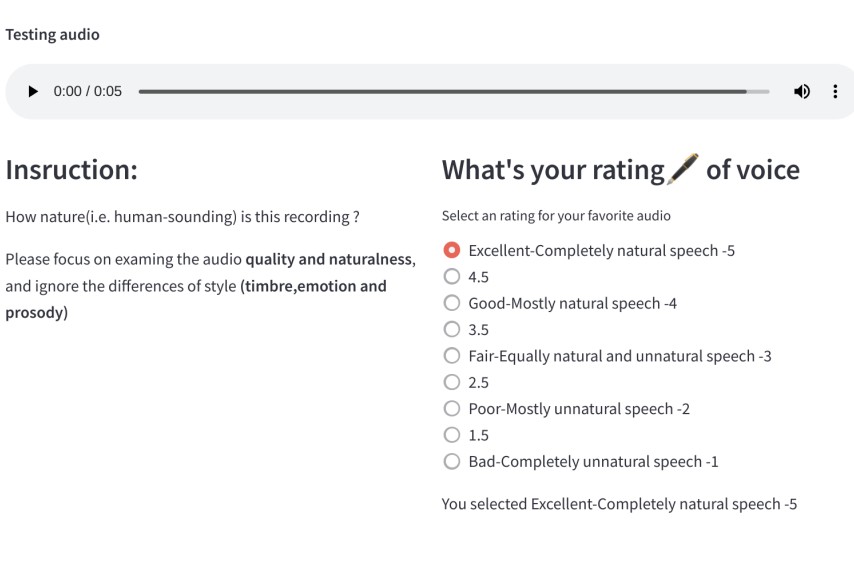

Figure 7: Screenshot of MOS test.

**Proposition 2.** (cf. §4.1): *Let $\boldsymbol{x}$ be the clean sample, $\boldsymbol{y}$ be it's corresponding noisy one, and $\boldsymbol{x}'$ be any neighbor from the neighbor set $\mathcal{S}(\boldsymbol{x})$. Then diffusion enhancement models can recover $\boldsymbol{x}$ with a high probability if the following inequality is satisfied:*

$$\log\left(\frac{p(\boldsymbol{x})}{p(\boldsymbol{x}')}\right) > \frac{1}{2\sigma_t^2}\left(\|\boldsymbol{x}-\boldsymbol{y}\|_2^2 - \|\boldsymbol{x}'-\boldsymbol{y}\|_2^2\right), \quad \forall \boldsymbol{x}' \in \mathcal{S}(\boldsymbol{x}), \tag{10}$$

*where $\sigma_t^2 = \frac{1-\bar{\alpha}_t}{\bar{\alpha}_t}$ is the variance of the Gaussian noise added at timestep $t$ in the forward diffusion process.*

*Proof.* The main idea is to prove that any point $\boldsymbol{x}'$ quite similar but different to the ground-true speech $\boldsymbol{x}$ should have a lower density than $\boldsymbol{x}$ in the conditional distribution so that the diffusion enhancement models can recover $\boldsymbol{x}$ with a high probability. In other words, we should have:

$$p(\boldsymbol{x}_0 = \boldsymbol{x}|\boldsymbol{x}_t = \boldsymbol{y}_t) > p(\boldsymbol{x}_0 = \boldsymbol{x}'|\boldsymbol{x}_t = \boldsymbol{y}_t) \tag{24}$$

According to Bayes' theorem, we have:

$$p(\boldsymbol{x}_0 = \boldsymbol{x}|\boldsymbol{x}_t = \boldsymbol{y}_t) = \frac{p(\boldsymbol{x}_0 = \boldsymbol{x}, \boldsymbol{x}_t = \boldsymbol{y}_t)}{p(\boldsymbol{x}_t = \boldsymbol{y}_t)} \tag{25}$$

$$= p(\boldsymbol{x}_0 = \boldsymbol{x}) \cdot \frac{p(\boldsymbol{x}_t = \boldsymbol{y}_t|\boldsymbol{x}_0 = \boldsymbol{x})}{p(\boldsymbol{x}_t = \boldsymbol{y}_t)}.$$

Applying Eq. (25) to Eq. (24), we obtain:

$$p(\boldsymbol{x}) \cdot \frac{1}{\sqrt{(2\pi\sigma_t^2)^d}} \exp\frac{-\|\boldsymbol{x}-\boldsymbol{y}\|_2^2}{2\sigma_t^2} > p(\boldsymbol{x}') \cdot \frac{1}{\sqrt{(2\pi\sigma_t^2)^d}} \exp\frac{-\|\boldsymbol{x}'-\boldsymbol{y}\|_2^2}{2\sigma_t^2} \tag{26}$$

$$\Leftrightarrow \log\left(\frac{p(\boldsymbol{x})}{p(\boldsymbol{x}')}\right) > \frac{1}{2\sigma_t^2}\left(\|\boldsymbol{x}-\boldsymbol{y}\|_2^2 - \|\boldsymbol{x}'-\boldsymbol{y}\|_2^2\right), \quad \forall \boldsymbol{x}' \in \mathcal{S}(\boldsymbol{x}),$$

and the proof is now complete. □

## A.2 Subjective evaluation

We conduct two types of Mean Opinion Score (MOS) tests to verify the quality of synthesized audio through human evaluation.

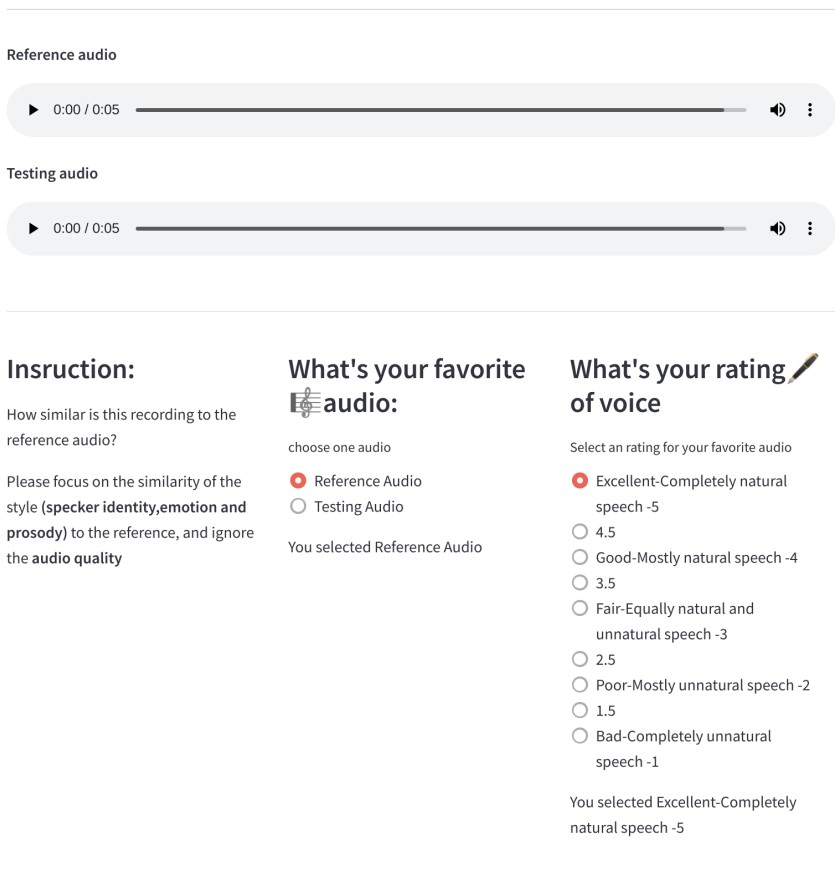

Figure 8: Screenshot of Similarity MOS test.

Table 2: MOS tests under different scenarios.

| Method | Scenarios | MOS↑ | Similarity MOS↑ | Scenarios | MOS↑ | Similarity MOS↑ |
|---|---|---|---|---|---|---|
| Unprocessed | | $3.60_{\pm0.31}$ | $3.33_{\pm0.34}$ | | $3.30_{\pm0.30}$ | $3.10_{\pm0.32}$ |
| DiffWave (dis) | | $3.80^{+0.20}_{\pm0.21}$ | $3.75^{+0.42}_{\pm0.24}$ | | $2.10^{-1.20}_{\pm0.28}$ | $1.00^{-2.10}_{\pm0.24}$ |
| DiffuSE | | $3.40^{-0.20}_{\pm0.20}$ | $4.17^{+1.84}_{\pm0.26}$ | | $3.00^{-0.30}_{\pm0.27}$ | $3.33^{+0.23}_{\pm0.21}$ |
| CDiffuSE | Matched | $3.85^{+0.25}_{\pm0.25}$ | $4.12^{+1.79}_{\pm0.31}$ | Mismatched | $2.55^{-0.75}_{\pm0.25}$ | $3.42^{+0.32}_{\pm0.29}$ |
| SGMSE | | $3.65^{+0.05}_{\pm0.28}$ | $3.95^{+0.62}_{\pm0.23}$ | | $2.83^{-0.47}_{\pm0.33}$ | $3.41^{+0.31}_{\pm0.25}$ |
| DR-DiffuSE | | $3.80^{+0.20}_{\pm0.25}$ | $3.84^{+0.51}_{\pm0.27}$ | | $2.48^{-0.82}_{\pm0.25}$ | $1.45^{-1.65}_{\pm0.33}$ |
| DOSE | | $4.05^{+0.45}_{\pm0.29}$ | $4.35^{+1.02}_{\pm0.21}$ | | $3.48^{+0.18}_{\pm0.26}$ | $3.17^{+0.07}_{\pm0.23}$ |

**Naturalness.** For audio quality evaluation, we conduct the MOS (mean opinion score) tests and explicitly instruct the raters to "focus on examining the audio quality and naturalness, and ignore the differences of style (timbre, emotion, and prosody)". The testers present and rate the samples, and each tester is asked to evaluate the subjective naturalness on a 1-5 Likert scale.

**Consistency.** For audio consistency evaluation, we explicitly instruct the raters to "focus on the similarity of the speech (content, timbre, emotion, and prosody) to the reference, and ignore the differences of audio quality". This is slightly different from the original definition of SMOS for speech synthesis. In the SMOS (similarity mean opinion score) tests, we pair each synthesized utterance with a ground truth utterance to evaluate how well the synthesized speech matches that of the target speaker. The testers present and rate the samples, and each tester is asked to evaluate the subjective consistency on a 1-5 Likert scale.

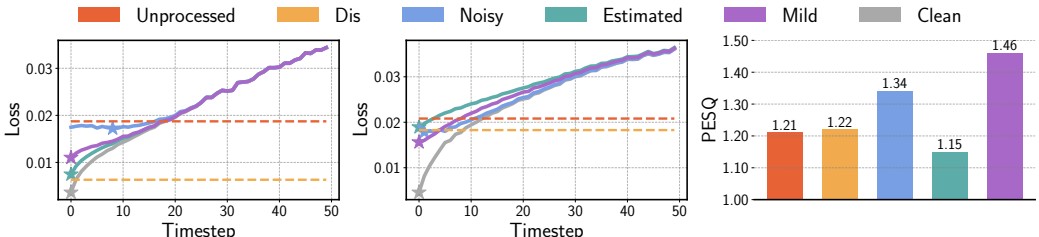

Figure 9: Performance of the unconditional diffusion enhancement model with adaptive prior. From left to right: (1) each step loss on matched VB; (2) each step loss on mismatched CHIME-4; (3) PESQ comparison for different priors on mismatched CHIME-4.

Our subjective evaluation tests are crowd-sourced and conducted by 15 volunteers. The screenshots of instructions for testers have been shown in Figure 7 and Figure 8. We paid $10 to participants hourly and totally spent about $300 on participant compensation.

The MOS results with the 95% confidence interval are shown in Table 2. we observe that: **(1)** Our method surpasses all baselines, demonstrating the strong ability of the proposed framework in synthesizing natural speech; **(2)** Our model can synthesize consistent speech to the golden speech, which is aligned with our motivation for algorithm design. It's also exciting to see that DOSE yields similar scores to methods with specific condition-injecting strategies (i.e., DiffuSE, CDiffuSE, and SGMSE) on Similarity MOS.

### A.3    Adaptive prior analysis

We now investigate the significance of the proposed adaptive prior (§4.1) and show why we need to use a mild version (Eq. (12)) of the condition factor.

We design three variants to investigate the effect of different condition optimizer settings on denoising performance. These variants are: (a) applying adaptive prior with the noisy speech; (b) applying adaptive prior with the estimated one (from the deterministic model); (c) applying adaptive prior with the mild condition (Eq. (12)). To control variables, we use an unsupervised diffusion model with adaptive priors. We conduct experiments on the matched VB dataset and mismatched CHIME-4 dataset respectively, and our results are shown in Figure 9.

As shown in Figure 9 (left), we plot the one-step loss on matched VB and obtain the following observations: **(1)** The trend of loss curves is in line with our analysis in §4.1 that we need to find a trade-off timestep for better performance. **(2)** Equipping the unconditional diffusion enhancement model with the adaptive prior technique has a certain denoising ability but is inferior to its counterpart discriminative model in matched scenarios. We attribute the second phenomenon to the limited denoising capacity of the unconditional diffusion enhancement models (cf. [9, 21]).

Although the performance of the unconditional diffusion enhancement model equipped with adaptive prior is mediocre in matched scenarios, as illustrated in Figure 9 (mid), it exhibits greater stability than discriminative models in mismatched scenarios. To verify the influence of different priors, we compare their PESQ on mismatched CHIME-4, shown in Figure 9 (right). We see that: **(1)** The deterministic model fails in mismatched scenarios and generates samples that are even worse than the unprocessed ones; **(2)** The diffusion enhancement model has strong generalizability and performs significantly better than the deterministic model; **(3)** The diffusion enhancement model loses its capability when using the estimated speech as the condition factor; and **(4)** Although the estimated speech is worse than the unprocessed one, the diffusion enhancement model equipped with a mild version of the condition factor achieves the best performance. This implies that the estimated speech can provide additional complementary guidance to the diffusion model, and the model can adaptively "separate the wheat from the chaff". Thus, using a mild version of the condition factor is important and necessary.

In summary, our research shows the strong generalizability of the diffusion enhancement model. However, we also find that the unconditional diffusion enhancement model has mediocre performance. This suggests that relying solely on the adaptive prior technique is not be sufficient, further emphasiz-

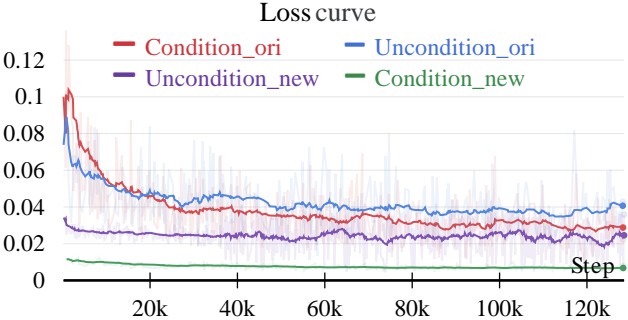

Figure 11: Investigation of the training objective. we compare the $\epsilon$-space and $x$-space loss curves.

ing the importance of the diffusion dropout operation – training conditional diffusion enhancement models in a supervised manner.

### A.4 Training objective investigation

We now examine the effect of our new training objective and demonstrate the necessity of using it.

Let's recall the sampling process of DOSE. In the first step, we generate a relatively good estimation of the clean speech. In the second step, we use DOSE with a small timestep to generate a better one. We plot the relationship between $\Delta\epsilon$ and $\Delta x_0$ in Figure 10. Specifically, we fix the $\Delta x_0$ as a constant and use Eq. (15) to calculate the corresponding $\Delta\epsilon$. This experiment aims to show how effort for calibration in x-space is equivalent to that in $\epsilon$-space. From Figure 10 we see that, as $t$ approaches zero, small changes in x-space amplify the implied prediction in $\epsilon$-space. There is a nearly 100-fold difference between the values of $\Delta\epsilon$ and $\Delta x_0$. This implies that the efforts of the diffusion enhancement model at small timesteps become negligible, causing diffusion models to lose their ability to recover natural-sounding speech from the defective speech estimated in the small timestep.

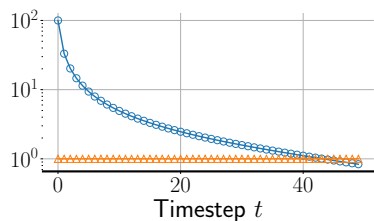

Figure 10: The relationship between $\Delta x_0 = |\hat{x}_0 - x_0|$ (orange) and $\Delta\epsilon = |\hat{\epsilon} - \epsilon|$ (blue).

We also plot the training loss curves in Figure 11. After substituting the training objective from $\epsilon$-space to x-space, we observe that the loss change becomes more significant. This demonstrates more effective participation of conditioning factors in diffusion training: training diffusion enhancement model with this new objective allows for easier and more effective exploitation of conditioning factors.

### A.5 Complexity analysis & why use two-steps generation?

In this subsection, we explain why we use two steps to generate speech.

**Why not one-step speech generation?** According to §4.4, we can generate speech in one shot using a trained conditional diffusion enhancement model to reduce error accumulation [21]. This one-step speech generation provides an appealing performance (Table 3). However, if we consider the diffusion model training as a multi-task paradigm, the denoising task at a smaller timestep $t$ is easier than that at a larger timestep [21]. Correspondingly, the primary estimation error occurs at the large timestep area and directly estimating clean speech at a large timestep will result in sub-optimal performance [28]. Meanwhile, we can't choose a small timestep $t$ as it will lead to the mismatched problem discussed in §4.1.

Since the conditional diffusion model can learn vital information from both the intermediate-generated and noisy speech – the estimated speech can provide complementary guidance to the diffusion model (Appendix A.3) – allowing us to further improve the result by generating speech with multiple steps.

Table 3: Ablation study for two-step speech generation.

| Method | Scenarios | PESQ↑ | STOI(%)↑ | Scenarios | PESQ↑ | STOI(%)↑ |
|---|---|---|---|---|---|---|
| Unprocessed | | 1.97 | 92.1 | | 1.21 | 71.5 |
| DOSE (fixed 1 step) | Matched | $2.47^{+0.50}_{\pm0.01}$ | $93.0^{+0.90}_{\pm0.05}$ | Mismatched | $1.38^{+0.17}_{\pm0.01}$ | $82.8^{+11.3}_{\pm0.05}$ |
| DOSE (handpicked 1 step) | | $2.50^{+0.53}_{\pm0.01}$ | $93.4^{+1.30}_{\pm0.05}$ | | $1.51^{+0.31}_{\pm0.01}$ | $86.4^{+15.7}_{\pm0.05}$ |
| DOSE (fixed 2 steps) | | $2.48^{+0.51}_{\pm0.01}$ | $93.1^{+1.00}_{\pm0.05}$ | | $1.44^{+0.23}_{\pm0.01}$ | $83.6^{+12.1}_{\pm0.05}$ |
| DOSE (handpicked 2 steps) | | $2.56^{+0.59}_{\pm0.01}$ | $93.6^{+1.50}_{\pm0.05}$ | | $1.52^{+0.32}_{\pm0.01}$ | $86.6^{+15.1}_{\pm0.05}$ |

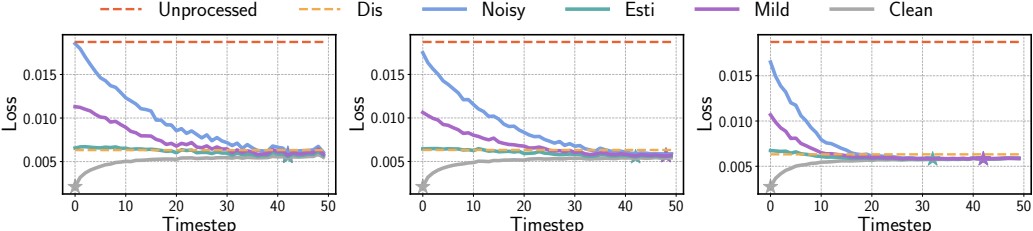

Figure 12: Performance of the conditional diffusion enhancement model with different dropout probability in VB. From left to right: (1) without dropout; (2) dropout 50% ; (3) dropout 90%.

**Hyper-parameter selection.** Basically, we need to set optimal hyper-parameters by evaluating the performance with a small batch of data. Suppose we choose the number of sampling steps as $K$ ($K < T$), and the amount of test data as $N$, then the computational complexity of the grid search is $\mathcal{O}\left(NT!/(T-K)!\right)$. Since $T$ is always set as a large number in the diffusion model's setup, the complexity introduced by choosing a large $K$ is often unmanageable.

**Trivial solution.** We have a simple alternative solution: defining the hyper-parameters empirically (e.g., equal intervals [7]). We present speech quality comparisons between empirically defined (fixed) and handpicked hyper-parameters in Table 3. As shown, the conditional diffusion enhancement model with handpicked hyper-parameters performs better than that with empirically defined hyper-parameters.

Although fixed hyper-parameters have inferior performance compared to handpicked ones, they still show appealing performance compared to prevailing diffusion enhancement baselines. Therefore, in situations where we can't evaluate the model in advance, we can use empirically defined hyper-parameters instead.

## A.6 Parameter sensitivity

There are two groups of hyper-parameters that are critical to model performance: the dropout probability $p$ for model training and two timesteps $\tau_1, \tau_2$ for model inference. In this subsection, we conduct parameter sensitivity experiments to investigate how these hyper-parameters affect the model's performance.

**Dropout probability $p$.** We vary the dropout probability $p$, setting it to $\{0.0, 0.5, 0.9, 1.0\}$, and plot the one-step loss on matched VB (Figure 12) and mismatched CHIME-4 (Figure 13) respectively. Please note that the figure with $p = 1$ has been excluded since it degenerates to a deterministic mapping-based model and the performance remains unchanged across all timesteps.

From Figure 12, we see that: **(1)** The proposed conditional diffusion enhancement model works. Compared to the deterministic mapping-based model, our model performs slightly better in matched scenarios and significantly better in mismatched scenarios; **(2)** When intermediate-generated speech $x_t$ is unreliable (large $t$), the condition factor $y$ plays a dominant role. **(3)** As the dropout probability increases, the model focuses more on the condition factor, while when the dropout probability is small, the loss curve oscillates when $t$ gets large.

From Figure 13, We find that: when using a higher dropout probability (such as $p = 0.5$ and $p = 0.9$), the model can generally achieve better generalizability. Note that if the dropout probability is set too high, the diffusion enhancement model will rely solely on the condition factor to estimate the clean

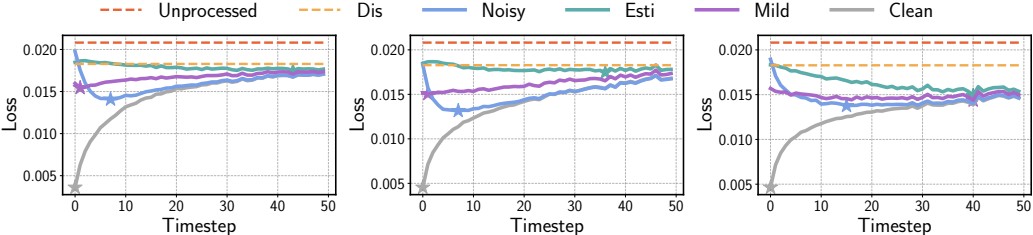

Figure 13: Performance of the conditional diffusion enhancement model with different dropout probability in CHIME-4. From left to right: (1) without dropout; (2) dropout 50% ; (3) dropout 90%.

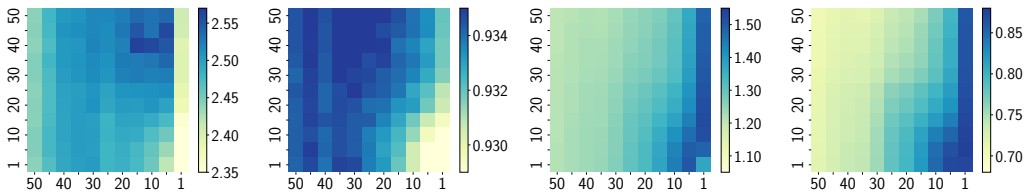

Figure 14: Performance of two-step speech generation. From left to right: (1) PESQ on matched VB dataset; (2) STOI on matched VB dataset; (3) PESQ on mismatched CHIME-4 dataset; (4) STOI on mismatched CHIME-4 dataset.

speech – and the diffusion enhancement model will degenerate into a deterministic model, losing its generalizability.

**Timesteps $\tau_1$ and $\tau_2$.** Considering the computational complexity of the "one-step" grid search, we search optimal hyperparameters with a slightly larger step. Specifically, we select both optimal $\tau_1$ and $\tau_2$ from the predefined set $\{1, 5, 10, 15, 20, 25, 30, 35, 40, 45, 50\}$. We show the PESQ and STOI performance of different combinations in Figure 14. We can observe that $\tau_1 > \tau_2$ (excluding STOI on matched VB dataset, and the difference is negligible: $0.930 \sim 0.934$) will lead to better performance, which is in line with our analysis (§4.4).

## A.7 Visual case of excessive suppression

We present several visual cases of excessive suppression.

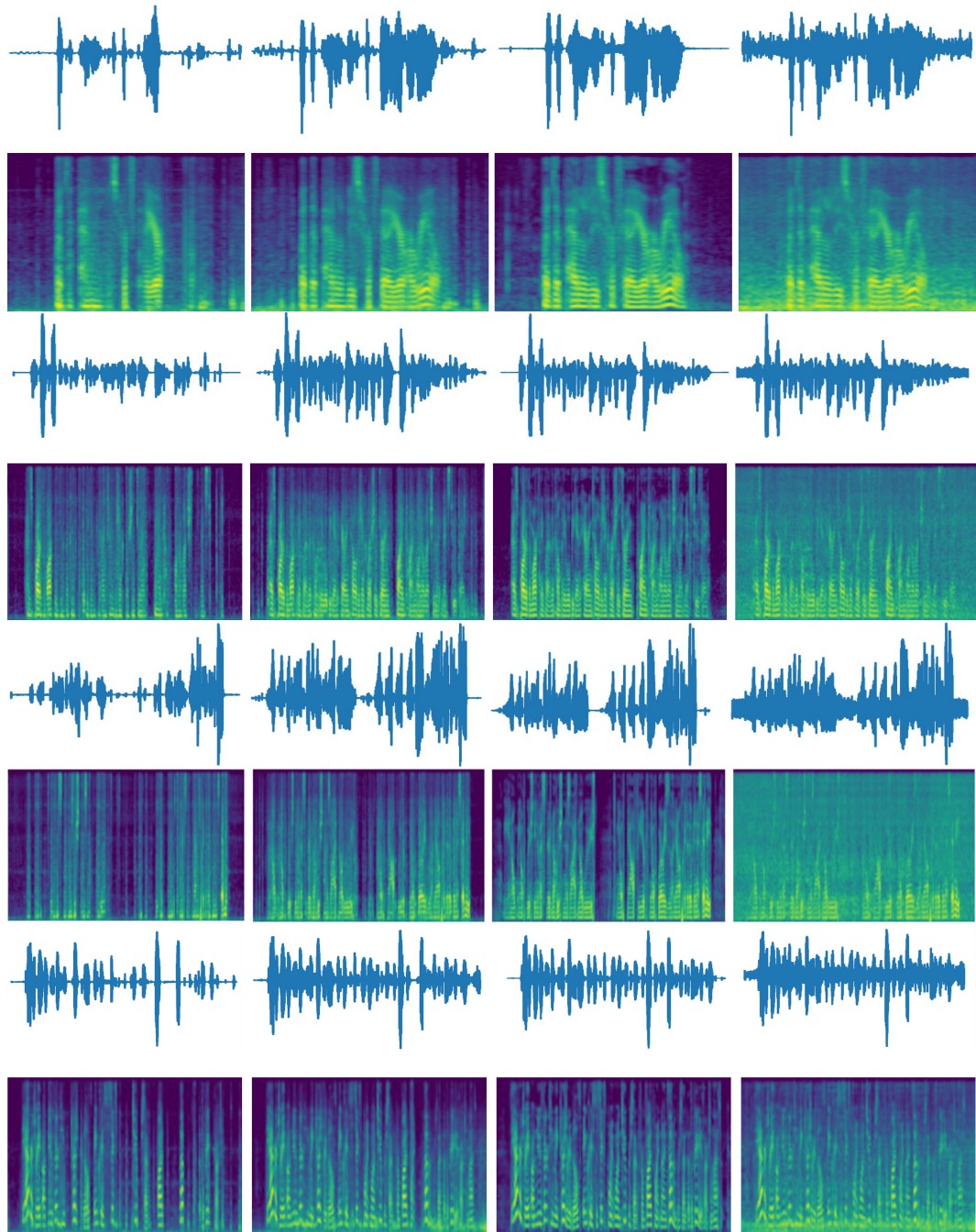

Figure 15: Excessive suppression visualization (CHIME-4, DOSE). From left to right: (1) estimated condition; (2) 2 steps; (3) clean; (4) noisy speech.

## A.8 Visual case of error accumulation

We present several visual cases of error accumulation.

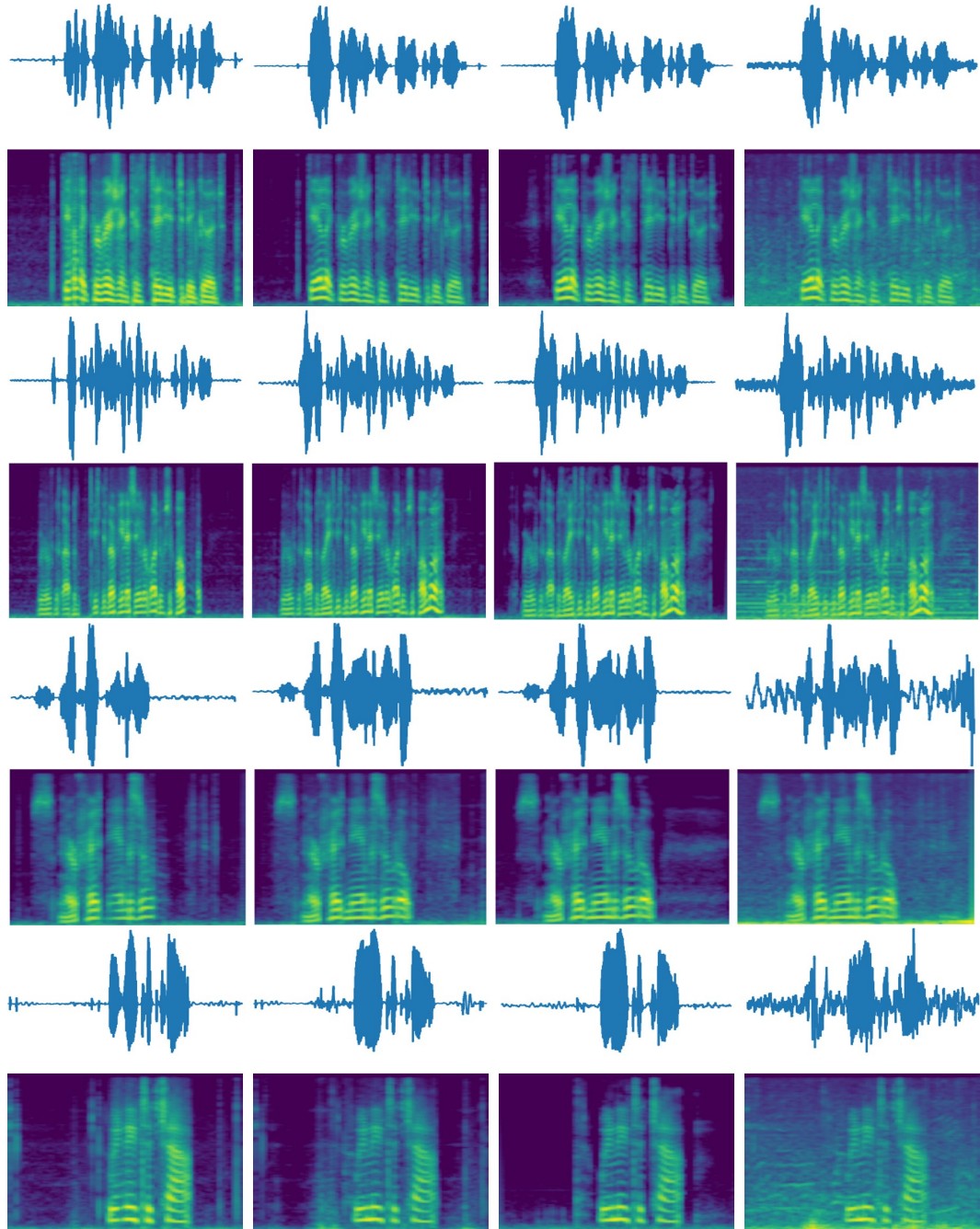

Figure 16: Error accumulation visualization (VB, DOSE). From left to right: (1) full (50) steps; (2) 2 steps; (3) clean; (4) noisy speech.

## A.9 Discussion

To help readers better understand our approach, we analyze the reasons behind the better generalizability of diffusion enhancement models compared to deterministic mapping-based models (from the robust training perspective), explain why we use 0.5 in the mild version of the condition factor, and discuss the broader impacts of speech enhancement methods.

### A.9.1 Generalization analysis

Given that full-step diffusion enhancement models have better generalizability than deterministic models, we now delve into the following question: if we are just using diffusion models as one-step (or two-step) denoisers, what accounts for their enhanced performance compared to deterministic mapping-based models?

To answer this question, we need to point out that training a diffusion model can be considered equivalent to training a multi-task paradigm. This involves training a model with shared parameters on multiple levels of Gaussian noise concurrently. Recent research [21] has demonstrated that the full training process of diffusion models leads to significantly improved one-shot denoising capabilities, which are more generalizable compared to previous works that trained standalone denoisers on a single noise level. Please refer to [21] (§5.2) for more details.

### A.9.2 Why use 0.5 in the mild version of the condition factor? (reviewer boQC)

Employing an equal weight provides stability, yet the performance during instances with low SNR would be compromised (albeit still superior to direct utilization of $y$ as a condition factor, unless the condition optimizer falters). One prospective solution involves introducing an additional adaptive strategy, i.e., $c = \alpha f_{\theta}(y) + (1 - \alpha)y$. We can design an adaptive *alpha* predictor and hope it can output $\alpha$ based on the quality of raw condition and samples from the condition optimizer. For example, when the condition optimizer produces lower-quality samples, giving more weight to the original condition factor would make sense. Conversely, if the raw condition factor has a low SNR, emphasizing the generated counterpart could be more effective. However, implementing this idea practically is intricate.

Given our strong reliance on diffusion-enhanced models to enhance generalization, any new adaptive strategy must be generalizable. For instance, training an adaptive alpha predictor on the (seen) VB-dataset (high SNR & consistent condition optimizer performance) could lead the model to consistently output higher $\alpha$ values for fusion. Unfortunately, this auxiliary model might not effectively adapt to variations when evaluating the mismatched (unseen) CHIME-4 dataset (low SNR & potential condition optimizer challenges). To this end, we might need other techniques such as data augmentation and adversarial training to improve its generalizability and robustness. This creates a dilemma: harnessing the speech diffusion model for overarching speech noise reduction generalization while simultaneously necessitating a pre-established generalized model to facilitate its implementation. So far, despite our efforts to train a strong alpha predictor, progress has been limited (the alpha predictor is still not generalizable, and the new system has no significant performance improvement over DOSE).

### A.9.3 Broader impacts

As speech enhancement technology continues to advance and become more prevalent, it's important to consider its broader impacts.

**Positive impacts.** The impact of speech enhancement technology on real-life situations, particularly for individuals with hearing impairments, cannot be overstated. Hearing aids have long been the primary solution for those with hearing loss, but they are not always effective in noisy environments or for certain types of hearing loss. Speech enhancement technology can greatly improve speech intelligibility and communication for hearing aid users. For example, some hearing aids have AI-powered speech enhancement that boosts speech quality.

In addition to the benefits for individuals with hearing impairments, speech enhancement technology also has significant implications for various applications. In transportation, clearer and more intelligible speech can improve communication between pilots and air traffic control, leading to safer and more efficient air travel. In industry, speech enhancement can improve communication on

noisy factory floors, leading to increased productivity and safety. In educational settings, speech enhancement can improve student comprehension and engagement during lectures and presentations.

**Negative impacts.** While speech enhancement technology has the potential to greatly improve communication and speech intelligibility, one potential concern is that speech enhancement could modify the semantic content of speech, potentially misleading listeners. Thus, it's important for developers of speech enhancement technology to consider this potential negative effect and work towards creating trustworthy systems.

### A.10 Experimental details

**Speech preprocessing.** We process the speech waveform at a 16 kHz sampling rate. To maintain dimensionality consistency within mini-batches, we pad each utterance to 2 seconds (32000 points) using a zero-padding technique.

**Basic architecture.** To make a fair comparison, we use DiffWave [7] as the basic architecture following [4, 9] – the only difference being the change in the way of condition-injecting since most speech enhancement methods will directly use noisy speech as the condition factor, rather than Mel-spectrogram. We concatenate the condition factor with the intermediate-generated sample along the channel dimension as the model's input. Specifically, the network is composed of 30 residual layers with residual channels 128. We use a bidirectional dilated convolution (Bi-DilConv) with kernel size 3 in each layer. We sum the skip connections from all residual layers. The total number of trainable parameters is 2.31M, slightly smaller than naive DiffWave (2.64M). Please refer to [7] and our code for more details.

