# OpenReview forum: "DOSE: Diffusion Dropout with Adaptive Prior for Speech Enhancement"
_NeurIPS.cc/2023/Conference — NeurIPS 2023 poster_

### Official Review · Reviewer_RTAK · 2023-06-11

**Soundness:** 3 good
**Presentation:** 3 good
**Contribution:** 3 good
**Rating:** 6
**Confidence:** 5

**Summary:**

This paper describes a new method for providing noisy-signal conditioning information (y) to the diffusion steps of a diffusion-based speech enhancement algorithm.  Three specific innovations are proposed: (1) improve dependence of x_0 on y by dropping out x_t, at random with Bernoulli probability p.  (2) In order to make innovation #3 possible, train each x_t explicitly using MSE of the implied x_0, rather than MSE of the error \epsilon.  (3) For greater efficiency, generate x_0 in only two steps, selected from the T-step trained diffusion process using validation data.

**Strengths:**

This paper proposes a diffusion-based speech enhancement with both improved performance and (because of the two-step inference) improved efficiency.  Both the performance gains and the efficiency gains are theoretically well motivated and empirically demonstrated.

**Weaknesses:**

Clarity:  (1) The dropout described in Eq. (13) is then not referenced again for the rest of the paper.  I think that's because Eq. (13) affects the T-step training process, while equations (14)-(18) are about the proposed reduction of inference from T steps to 2 steps.  But the division into training and testing is never really made explicit.  I think this is because the derivations assume that the reader has fully understood Figure 1, but Figure 1 cannot be fully understood until one has first understood the algorithm; I had to go back and examine Figure 1 after reading the derivations in order to know what's going on.  (2) I think that Eq. (17) should have an integral over dx_\tau.  The algorithm might use the one-point approximation, but you've done a great job up to this point of keeping the theoretically-required integral in your equations, it seems a shame to abandon it here.  (3) Shouldn't the first term on the RHS in Eq. (18) be p_\theta(\hat{x}_{\tau_2}|...)?  Or are you trying to say that p(x_{\tau_2}|...) = p(\hat{x}_0|...)?  That doesn't seem correct, since it misses the interpolation step.  ... also, I think there should be an integration over d\hat{x}_{\tau_2}.

**Questions:**

Significance:  Dropout enhances accuracy.  The two-step inference enhances efficiency.  Is there any interaction between these two things?  It seems like the dropout might reduce the accuracy degradation that two-step inference would otherwise incur; is that true?  Similarly, why two -- is two the optimum number of steps in any way?  Results show pretty clearly that two is better than one, but is there any theoretical reason for that?  The theory seems to predict simply that the more steps you have, the more accurate is the inference.

"Diffusion enhancement methods have better generalizability than deterministic methods" -- By "deterministic methods" I think you mean DiffWave.  In what sense is that a deterministic method?

p. 6 Considering the equivalently -> Considering the equivalence

p. 8 We contribute -> We attribute



**Limitations:**

Limitations and ethical considerations are not explicitly addressed.

---

> ### Author Rebuttal · Authors · 2023-08-07
>
> Thank you for the detailed review and thoughtful feedback. Below we address specific questions.
> ***
> >Q1: The dropout described in Eq. (13) is then not referenced again for the rest of the paper. I think that's because Eq. (13) affects the $T$-step training process, while equations (14)-(18) are about the proposed reduction of inference from $T$ steps to 2 steps. But the division into training and testing is never really made explicit. I think this is because the derivations assume that the reader has fully understood Figure 1, but Figure 1 cannot be fully understood until one has first understood the algorithm; I had to go back and examine Figure 1 after reading the derivations in order to know what's going on.
>
> Thank you for your valuable feedback and comments! We apologize for any confusion caused by the paper writing. We will move Figure 1 to Sec 4 and reorganize Sec 4.2 -- explicitly divide training and testing as your suggested.
> ***
> >Q2: I think that Eq. (17) should have an integral over $dx_{\tau}$. The algorithm might use the one-point approximation, but you've done a great job up to this point of keeping the theoretically-required integral in your equations, it seems a shame to abandon it here.
>
> You are right that Eq. 17 should have an integral over $dx_{\tau}$. We will correct these in the revision.
> ***
> > Q3: Shouldn't the first term on the RHS in Eq. (18) be $p\_\theta(\hat{x}\_{\tau\_2}|...)$? Or are you trying to say that $p(x\_{\tau\_2}|...) = p(\hat{x}\_0|...)$? That doesn't seem correct, since it misses the interpolation step. ... also, I think there should be an integration over $d\hat{x}\_{\tau\_2}$.
>
> Thank you for pointing out these oversights in Eq. 18! The RHS in Eq. 18 should be $p\_\theta(\hat{x}\_{\tau\_2}|...)$ and it should be an integration over $d\hat{x}\_{\tau_2}$. We will correct these in the revision.
>
> >Q4: Significance: Dropout enhances accuracy. The two-step inference enhances efficiency. Is there any interaction between these two things? It seems like the dropout might reduce the accuracy degradation that two-step inference would otherwise incur; is that true? Similarly, why two -- is two the optimum number of steps in any way? Results show pretty clearly that two is better than one, but is there any theoretical reason for that? The theory seems to predict simply that the more steps you have, the more accurate is the inference.
>
> Great question! It's important to clarify that full-step generation doesn't always yield better results compared to few-step generation. This phenomenon was exemplified in DiffuSE [5], where generating speech in 6 steps outperforms 50-step generation. Similarly, in [6, 7], authors demonstrated that generating samples in 1/10 steps performs better than full-step image generation (purification).
>
> The notion that "the more steps, the more accurate the inference" holds true only if each step produces a better estimation than the preceding one. Now, suppose the model can always generate a better condition factor. According to Proposition 1, a smaller $t$ can always be chosen. From a holistic standpoint, this loop continues as each step generates a better condition factor, thus the more steps, the more accurate the result (Sec 4.1, lines 164-166). However, in practice, this ideal scenario is hindered by empirical and generalization errors. Various factors contribute to model errors, including complexity, architecture, data quality, optimization, and stochasticity. Empirical evidence [16, 17] indicates that 2-step generation tends to outperform 1-step generation. This is because model can always generate improved estimates compared to the initial condition $c=y$, making 2-step better than 1-step in most cases [16, 17]. However, we cannot guarantee that $K$-step ($K > 2$) is better than 2-step. We showed multiple visual cases in Fig. 5 and Fig. 15 that increasing the number of sampling steps will lead to inconsistency (error accumulation) problem and subpar results. Additional results for different $K$ (i.e., 2, 6, 50 steps) are attached in the table (in General Response). Note that research on speech enhancement based on progressive learning [18, 19] also shows that iterative learning over 5 steps often leads to performance degradation. Taking into account the computational complexity of optimizing hyperparameters for $K$ (Appendix A.6, lines 571-575), we opt to directly set $K=2$ for both efficiency and stability reasons. We appreciate your question and hope this explanation clarifies our approach and reasoning.
> ***
> >Q5: "Diffusion enhancement methods have better generalizability than deterministic methods" -- By "deterministic methods" I think you mean DiffWave. In what sense is that a deterministic method?
>
> To ensure a fair comparison, we kept the model architecture exactly the same as that of the DiffWave, but used $y$ as the $x_t$ (so the model's input is two noisy speech $y$ concatenated along the channel dimension). We used a zero vector as the time step embedding so that it did not contain any additional information. This common practice of comparing generative diffusion models with deterministic counterparts is prevalent in the literature [13, 20].
> ***
> >Q6: p. 6 and p. 8 typos.
>
> Thanks for spotting the typos! We will correct these in the revision.

---

### Official Review · Reviewer_k89b · 2023-06-26

**Soundness:** 3 good
**Presentation:** 3 good
**Contribution:** 3 good
**Rating:** 7
**Confidence:** 4

**Summary:**

This paper presents a solution to the problem of condition-collapse in denoising diffusion models for speech enhancement by introducing the adaptive prior and sample dropout techniques. The paper is well-written and provides valuable insights into the functioning of the denoising diffusion probabilistic model for speech enhancement. While the concept of adaptive priors for conditioning the generative process is not entirely novel and has been explored in vision-related tasks, the authors' application of this technique, along with theoretical analysis, is intriguing. The authors also justify their choices regarding noise scheduling and propose a faster method for sampling clean speech from the trained model based on intermediate approximation.

The experimental strategy adopted in this study assesses the generalization capability of the proposed model using objective metrics such as STOI and PESQ, as well as subjective scores like CBAK and COVL. The effectiveness of the mixed conditioning strategy is demonstrated through the analysis of spectrogram plots, which is an interesting observation. It is important to note that while the proposed technique may not consistently outperform baseline methods across all scenarios, it does excel in specific matched scenarios.

**Strengths:**

In this paper, a novel approach is introduced to address the problem of condition collapse in diffusion models for speech enhancement. The authors propose the utilization of adaptive prior and sample dropout techniques, which offer an interesting and promising solution to this issue. Furthermore, the paper delves into the theoretical aspects of clean speech recovery, shedding light on the conditions and constraints necessary for successful restoration.

One notable contribution of this work is the development of a fast sampling technique, which not only proves effective in the context of speech enhancement but also holds potential for application in other conditional generation tasks. This aspect highlights the broader implications and versatility of the proposed approach.

To evaluate the efficacy of the proposed technique, the authors conduct comprehensive experiments and compare their approach against various diffusion-based models specifically designed for speech enhancement. The experiments are meticulously designed and executed, providing a thorough analysis of the results. This level of detail and scrutiny enhances the credibility of the proposed approach and contributes to a better understanding of its strengths and limitations.

**Weaknesses:**

One main weakness in my opinion is the understanding of Proposition 2. I do not understand how a diffusion model has high probability of recovering ground-truth if the inequality 23 from appendix holds. I might be missing some theoretical analysis on diffusion models but I am giving the benefit of doubt to the authors.

The dataset section lacks the details about the type and level of noise present in Chime and Voicebank corpora. Another issue is in the experiment section where the authors have shown impressive performance on a wide-range of metrics. I believe that WER can be easily calculated in the evaluation and is a very straight-forward way to compare the noise-reduction performance.

**Questions:**

None

**Limitations:**

The conclusion section mentions that the model is sensitive to the choice of dropout probability and the sampling time-indices.

---

> ### Author Rebuttal · Authors · 2023-08-07
>
> Thank you for the detailed review and thoughtful feedback. Below we address specific questions.
> ***
> >Q1: One main weakness in my opinion is the understanding of Proposition 2. I do not understand how a diffusion model has high probability of recovering ground-truth if the inequality 23 from appendix holds. I might be missing some theoretical analysis on diffusion models but I am giving the benefit of doubt to the authors.
>
> Unlike directly maximizing the condition probability $p(x_0 = x|x_t = y_t)$ or the difference between the target speech and candidates $||p(x_0 = x|x_t = y_t) - \max (p(x_0 = x^{\prime}|x_t = y_t); x^{\prime} \in \mathcal{S}(x))||_2^2$, Proposition 2 (Eq. 23) presents a relatively relaxed constraint. If Proposition 2 holds, considering the characteristics of unconditional diffusion models / score-based models [14, 15], particles starting at the adaptive prior $y_t$  are more likely to converge to the ground-truth objective $x$ through an iterative MCMC procedure (known as Langevin dynamics), rather than other natural but inconsistent candidates $\forall x^{\prime} \in \mathcal{S}(x)$. This constraint also suggests that we should select a smaller $t$ and narrow the gap between the condition factor and $x_t$, which provides guidance for the subsequent DOSE design. We appreciate your question and hope this explanation addresses your concerns.
> ***
> >Q2: The dataset section lacks the details about the type and level of noise present in Chime and Voicebank corpora. Another issue is in the experiment section where the authors have shown impressive performance on a wide-range of metrics. I believe that WER can be easily calculated in the evaluation and is a very straight-forward way to compare the noise-reduction performance.
>
> Thanks for the suggestion! The VoiceBank-DEMAND dataset is a classical benchmark dataset for speech enhancement using clean speech from the VCTK corpus. The training utterances are artificially contaminated with eight real-recorded noise samples from the DEMAND database and two artificially generated noise samples (babble and speech shaped) at 0, 5, 10, and 15 dB SNR levels, amounting to 11,572 utterances. The testing utterances are mixed with different noise samples at 2.5, 7.5, 12.5, and 17.5 dB SNR levels, amounting to 824 utterances in total. The CHiME-4 simulated test data is created based on real-recorded noises from four real-world environments (including street, pedestrian areas, cafeteria and bus) based on four speakers, with a total of 1320 utterances. Following [12], we use the signals from the fifth microphone for evaluation. We will update the dataset details in the revision.
>
> We have evaluated all speech enhancement methods using two public pre-trained ASR models (CRDNN-RNNLM and Conformer-Transducer) from huggingface. The result is shown in the table below.
>
> |  |  |  | VoiceBank |  |  |  |  |
> |---|:---:|:---:|:---:|:---:|:---:|:---:|:---: |
> | Model | DOSE | DiffuSE | CDiffuSE | SGMSE | DR-DiffuSE | DiffWave(dis) |
> | CRDNN-RNNLM | **12.77%** | 14.28% | 12.97% | 14.81% | 13.01% | 14.31% |
> | Conformer-Transducer | **9.83%** | 10.83% | 9.96% | 11.67% | 9.89% | 10.90% |
>
> |  |  |  | CHIME-4 |  |  |  |  |
> |---|:---:|:---:|:---:|:---:|:---:|:---:|:---: |
> | Model | DOSE | DiffuSE | CDiffuSE | SGMSE | DR-DiffuSE | DiffWave(dis) |
> | CRDNN-RNNLM | 39.66% | 38.10% | 37.59% | **37.17%** | 44.51% | 71.92% |
> |  Conformer-Transducer | 30.30% | 28.44% | **28.41%** | 28.62% | 31.07% | 59.76% |
>
>
> We have the following observations:
>
> * On the VoiceBank-DEMAND dataset (matched scenario), the performance gap between diffusion enhancement models and deterministic model is not prominent. For instance, WER ranges from 0.127 to 0.148 with CRDNN and from 0.098 to 0.116 with Conformer-Transducer.
>
> * On the CHIME-4 dataset (mismatched scenario), diffusion enhancement models significantly outperform deterministic models in terms of performance, e.g., WER of diffusion enhancement models ranges from 0.371 to 0.445 with CRDNN and from 0.284 to 0.310 with Conformer-Transducer, while WER of the deterministic model is  0.719 with CRDNN and 0.597 with Conformer-Transducer.
>
> * We find our method has no significant differences with DiffuSE, CDiffuSE, and SGMSE on WER evaluation.

---

> > ### Comment · Reviewer_k89b · 2023-08-20
> > **No additional questions.**
> >
> > I thank authors for addressing my questions. I believe an accept (7) is a good score for this paper.

---

### Official Review · Reviewer_boQC · 2023-07-05

**Soundness:** 3 good
**Presentation:** 2 fair
**Contribution:** 2 fair
**Rating:** 6
**Confidence:** 5

**Summary:**

This paper focuses on a new approach in the field of speech enhancement called DOSE, which effectively addresses the problem of conditional collapse by incorporating conditional information into a diffusion enhancement model.DOSE employs two effective conditional enhancement techniques that can significantly improve the performance of the model while ensuring its efficiency. The paper demonstrates the efficiency and effectiveness of the method through comprehensive experiments on a benchmark dataset.

**Strengths:**

1. In this paper, the authors propose an Adaptive Prior, aimed at incorporating conditioning information during the generation process, thereby ensuring greater consistency in the generated samples and augmenting the efficacy of speech enhancement.

2. The paper elucidates that by employing dropout operations during the training phase, the model is compelled to prioritize conditioning elements, which efficaciously mitigates the conditioning collapse issue. This methodology engenders a dependency on conditioning information within the model during generation, culminating in the synthesis of more coherent speech.

3. The authors undertake a comparative analysis between DOSE and extant diffusion-based speech enhancement models on two public datasets. Notably, DOSE attains superior performance with an exceedingly limited number of sampling steps, which substantiates the efficacy of the proposed method.

**Weaknesses:**

This work realizes a two-step sampling process. The designs in sampling process include:

1. The parameter $T_1$ denotes an intersection point, enabling a shallow reverse process by leveraging the noisy speech y. Similar mechanism can be visited in text-to-speech synthesis, DiffSinger (AAAI, 2022) and image editing, SDEdit (ICLR 2022).
2. The coarse estimation $\hat{x}_{0}$ denoised from $T_1$ is first mixed with y as adaptive prior,  and then corrupted with a shallow forward process, which generates the latent representation at $T_2$.
3. The high-quality estimation $\hat{x}_{0}$ can recovered from $T_2$ in one-step.

Two designs in training include:

1. The estimation target in training objective is set as clean waveform instead of noise.
2. The $x_t$ is randomly dropped out to force the model to rely on the conditioning information $y$.

Questions:

1. I can understand the dropout operation in training is helpful to utilizing the conditioner y. However, the adaptive prior is used to obtain the latent representation at T_2. I think it is manipulating the sampling trajectory. What is the relationship with condition optimizer? The unchanged noisy observation y has been provided as the condition. The abstract claims two condition-augmentation techniques.
2. I do not understand the comparison study of adaptive prior analysis very well. What are the three variants when computing the adaptive prior \hat{x}_{0}? Why is an unconditional diffusion model used? To compare the design of Eq. 12, the conditional generation should be fixed.
3. This work mentions the error accumulation of diffusion models. However, the high-quality generation of diffusion models is usually guaranteed by its iterative refinement mechanism. In this work, 50 time steps are used in training, while 2 steps are used in sampling. I am curious about the results of increasing the number of sampling steps.

Experiments:

1. Subjective tests MOS and SMOS are conducted. But where is the demo page showing the generated samples?
2. DiffWave does not claim one-step mapping for either waveform generation or denoising. Is it good to use it as a discriminative model?
3. Generative baseline models such as DiffuSE, CDiffuSE, and SGMSE have been changed to keep the uniform model architecture and training method with this work. Will this cause performance decrease in their methods? What are the results of those unchanged baseline models?

Others:

1. The thesis writing looks over-complicated. Moreover, the adaptive prior does not mean condition optimizer from my perspective. It is manipulating the sampling trajectory with observation y.
2. I suggest showing the training and sampling algorithms in the main content instead of in appendix.

**Questions:**

My detailed questions are as described above.

**Limitations:**

There are limitations to its use in real-time scenarios.

---

> ### Author Rebuttal · Authors · 2023-08-07
>
> Thank you for the detailed review and thoughtful feedback. Below we address specific questions.
> ***
> >Q1: I think adaptive prior is manipulating the sampling trajectory. What is the relationship with condition optimizer? The unchanged noisy observation $y$ has been provided as the condition. The abstract claims two condition-augmentation techniques.
>
> Yes, you are right, our adaptive prior is designed to explicitly incorporate condition knowledge by manipulating the sampling trajectory. Ideally, we can directly use the noisy speech $y$ to generate the adaptive prior like SDEdit [2]. However, when in low-SNR scenarios (speech signal is contaminated by noises severely), we have to choose a relatively large $t$ to guarantee an acceptable error bound (Proposition 1). According to Proposition 1 and 2, if we can narrow the gap between the condition factor and input $x$, we can opt for a smaller (better) $t$. This enables the model to prevent excessive removal of original semantic information in the condition factor (line 168-172). To this end, we can employ a condition optimizer to generate an enhanced adaptive prior (Eq. 12). And the second condition-augmentation technique involves employing a condition optimizer to generate an adaptive prior and explicitly injecting the condition knowledge at the inference stage. On the whole, it is a condition augmentation technique tailored for diffusion enhancement models,  aiming at alleviating the condition collapse problem. We appreciate your question and hope this explanation addresses your concerns.
> ***
> >Q2: What are the three variants when computing the adaptive prior $\hat{x}_0$? Why is an unconditional diffusion model used? To compare the design of Eq. 12, the conditional generation should be fixed.
>
> We explored using unconditional diffusion model with adaptive prior technique as a first attempt to address the condition collapse problem of conditional diffusion enhancement methods (Sec 4.1). We found it essential to consider failure cases of the condition optimizer, particularly in mismatched scenarios -- using the estimated speech directly from the condition optimizer (Eq. 11, similar to DifFace [3] and DiffSinger [4]) could lead to excessive suppression problem. To investigate this further, we defined three variants (cf. Appendix A.4, lines 504-506):
> * Applying the adaptive prior with the noisy speech (similar to SDEdit [2]);
> * Applying the adaptive prior with the estimated speech (similar to DifFace and DiffSinger, Eq. 11);
> * Applying the adaptive prior with a milder one (Eq. 12).
>
> In our experiments (reported in Appendix A.4), we discovered that the mild condition is more stable in complex scenarios, while the unconditional diffusion model showed limited effectiveness in matched scenarios. Both of these insights were very valuable and used for designing DOSE (cf. Sec 4.2).
> ***
> >Q3: This work mentions the error accumulation of diffusion models. However, the high-quality generation of diffusion models is usually guaranteed by its iterative refinement mechanism. In this work, 50 time steps are used in training, while 2 steps are used in sampling. I am curious about the results of increasing the number of sampling steps.
>
> Great question! We would like to emphasize that the definition of high-quality in image/speech synthesis is not consistent with the one in speech enhancement: the former focus on naturalness, while the latter focus on point-to-point consistency. We presented multiple visual cases in Fig. 5 and Fig. 15 which illustrate that increasing the number of sampling steps will lead to inconsistency (error accumulation) problem and subpar results. Additional experimental results are shown in the above table (in General Response). Note that several recent works [5, 6, 7] share the same findings as ours that the consistency gets worse when starting with a large $t$.
> ***
> >Q4: Subjective tests MOS and SMOS are conducted. But where is the demo page showing the generated samples?
>
> We have made the test page public. Due to the NIPS policy that "rebuttal should not contain any links to external pages", we have send an anonymized link to the AC in a separate comment.
> ***
> >Q5: DiffWave does not claim one-step mapping for either waveform generation or denoising. Is it good to use it as a discriminative model?
>
> We use DiffWave as a discriminative model due to the following three reasons.
>
> * As stated in DiffWave [8], their network architecture is based on WaveNet [9], a speech synthesis model that has been successfully applied to **speech enhancement** [10] and separation [11];
>
> * Pioneer works [5, 12] in DDPM-based speech enhancement are all based on DiffWave. To ensure a fair comparison, we need to keep the model architecture exactly the same;
>
> * There exists a concurrent work [13] that does a similar thing to ours, i.e., they use NCSN++ [14], another generative model as the basic architecture.
> ***
> >Q6: Generative baseline models such as DiffuSE, CDiffuSE, and SGMSE have been changed to keep the uniform model architecture and training method with this work. Will this cause performance decrease in their methods? What are the results of those unchanged baseline models?
>
> As we explained in Appendix A.12 (line 641-644), current SOTA speech enhancement methods directly use noisy speech as the condition factor, rather than Mel-spectrogram. We note that this slight modification also leads to an improved performance of these methods (please see the respective reported performance for more details).
> ***
> >Q7: There are limitations to its use in real-time scenarios.
>
> We would like to emphasize that one of the attractive properties of our method is that speech can be generated in only 2 steps. We believe our method can shed light on the design of future fast diffusion enhancement models.
> ***
> >Q8: Writing and paper organization advice.
>
> Thanks for the suggestion! We will reorganize the paper in the revision.

---

> > ### Comment · Reviewer_boQC · 2023-08-18
> > **Discussion**
> >
> > Thank you for your detailed rebuttal and explanations provided to address my concerns. I have read through your answers and would like to further emphasize and elaborate on some points:
> >
> > **A new question**: may I ask the main target of this work? Are the improving techniques designed for achieving high-generation quality or fast sampling speed?
> >
> > **Regarding Q1**:
> >
> > OK, I understand that condition optimizer means the milder adaptive prior shown in Eq. 12. You inject scaled observation y into the latent representation at the second sampling step \hat x_t2. I was hoping to ask two questions about the milder prior:
> >
> > 1. You set the equal weight (predefined as 0.5) for the denoising result of the first sampling step f_\theta(y_{t1}, y, t1) and the observation y. Does this mean that you have equal confidence for these two terms, although the observation y may have different signal-to-noise (SNR)?
> >
> > 2. When the observation y has a low SNR, would it be helpful to consider it as Eq. 12? If y is very noisy, you still inject it into the denoising result of the first sampling step. Would it be informative to the final generation results?
> >
> > **Regarding Q3**: You mention that increasing the number of function evaluations (NFEs) will lead to inconsistency (error accumulation) and less satisfactory results when compared to the two-step approach. If this is indeed the case, it raises a fundamental question: what is the motivation or benefit of utilizing the diffusion-based framework for this task?
> >
> > From my perspective, the intrinsic value of diffusion models lies in their promising generation quality achieved by iterative sampling. Moreover, a trade-off between (controlled) generation quality and sampling speed could be achieved by tuning NFEs.  If the generation process has been limited to NFE=1 or NFE=2 because of the error caused by discretized sampling step, I think the method should be compared with more discriminative models.
> >
> > **Regarding Q5**:
> > Regarding using DiffWave as the baseline of discriminative models: I am not claiming that the WaveNet architecture is not good. But I do not believe that changing DiffWave to one-step mapping is a proper choice of discriminative models. Other papers like CDiffuSE, UNIVERSE, and SGMSE+ show the comparison results with several published discriminative methods.
> >
> > Comparing with such deterministic models would not only bolster the credibility of your results but also provide readers with a clearer context of where DOSE stands in terms of performance within the broader speech enhancement landscape.
> >
> > I look forward to your further clarifications on these matters.

---

> > > ### Author Response · Authors · 2023-08-19
> > > **(1/2) Response to Further Questions Raised by Reviewer boQC**
> > >
> > > We appreciate your valuable feedback! Below we answer your questions :)
> > > ***
> > > > Q1: The main target of this work.
> > >
> > > Our aim is to tackle the conditional collapse issue [1] in the conditional diffusion enhancement model, ultimately improving denoising performance. We present a model-agnostic approach equipped with two innovative conditional augmentation strategies to effectively exploit condition knowledge. Our adaptive prior bolsters inference speed by shortening the sampling trajectory from $T$-step to several steps. Considering the error accumulation problem, we set $K=2$, further enhance the inference speed. We'd like to stress that 2-step is not the optimal choice -- for both efficiency and stability reasons (please see Reviewer RTAK, R4 for more details).
> > > ***
> > > > Q2: (1) Why set equal weights? \& (2) The influence of injecting $y$ when it has a low SNR.
> > >
> > > (1): We'd like to emphasize that the motivation behind our strategy (Eq. 12) stands apart from the concept of "confidence''. Instead, it is more like a simple residual layer that integrates raw information to circumvent excessive suppression problem. From another perspective, in cases where the performance of the conditional optimizer is uncertain—whether it performs well or not—a judicious and logical way is to opt for a merging value of 0.5. This decision finds its roots in the principles of the Maximum Entropy Principle.
> > >
> > > (2): Low-SNR $y$ can impede the effectiveness of the adaptive prior mechanism. This is because when dealing with a low-SNR condition factor , it becomes necessary to select a relatively large value for $\tau$ to satisfy the Proposition 1 -- the original semantic information will also be removed if $\tau$ is too large (cf. Sec 4.1, line 162-167). However, it's important to note that establishing the condition prior "adaptively'' is one of the most attractive properties of adaptive prior mechanisms. When the signal-to-noise ratio of the condition factor is high, the advantages of the adaptive mechanism will be fully revealed, as it can provide an informative prior and shorten the sampling path. Even with a low-SNR $y$, in instances where the condition optimizer is effective, there's potential to attain an improved condition factor (as defined in Eq. 12) compared to the straightforward utilization of $y$.
> > > ***
> > > > Q3: (1) Motivation behind diffusion enhancement models \& (2) The intrinsic value of diffusion models (iterative sampling).
> > >
> > > (1): Almost all diffusion enhancement works claim that their methods generalize better than deterministic models.
> > >
> > > (2): Good question! We concur with your insight that more steps always lead better sample quality (for generation tasks). We'd like to stress that when applying DDPMs to fine-grained point-to-point mapping (regression) tasks, full-step generation doesn't always yield better results compared to few-step generation. Not only our experiments can verify this (Figure 5 and Figure 15 for visualization \& quantitative results in General Response), but also the recent works in speech enhancement [2] (6-step is better than 50-step), inverse problem [3] (20-step is slightly better than 100-step) , and image purification [4, 5] (1/10-step is better than full-step). Additionally, we advocate that the specific training paradigm of DDPM also brings benefits. We presented a generalization analysis (see Appendix A.11) explaining why diffusion enhancement models exhibit superior generalizability over deterministic counterparts (from the perspective of multi-task training). Recent research [4] highlights that the comprehensive training process of diffusion models substantially enhances one-shot denoising capabilities, making them more adaptable compared to previous works that focused on standalone denoisers at a single noise level.
> > > ***
> > > > Q4: More discriminative methods should be compared like CDiffuSE, UNIVERSE, and SGMSE+.
> > >
> > > We'd like to emphasize that our approach is a model-agnostic solution (claimed in our Abstract) to tackle the condition collapse issue. Unlike works such as CDiffuSE, UNIVERSE, and SGMSE+ that focus on designing exceptional network architectures for achieving state-of-the-art performance, we take a different path. Our work aligns more closely with a concurrent study [6] (published at ICASSP 2023), which compares the diffusion enhancement model to a discriminatively trained neural network, employing the same network architecture for restoration tasks. (As they suggested: "However, to make a fair comparison of these two conceptually different approaches, similar network architectures and same training data should be used.'') In the future, we will adapt our method to more popular SE methods as your suggestion.
> > > ***
> > > We appreciate your questions and hope these responses addresses your concerns.

---

> > > > ### Author Response · Authors · 2023-08-19
> > > > **(2/2) References**
> > > >
> > > > [1] Revisiting Denoising Diffusion Probabilistic Models for Speech Enhancement: Condition Collapse, Efficiency and Refinement, AAAI, 2023.
> > > >
> > > > [2] A Study on Speech Enhancement Based on Diffusion Probabilistic Model, APSIPA, 2021.
> > > >
> > > > [3] Denoising Diffusion Restoration Models, NeurIPS, 2022.
> > > >
> > > > [4] (Certified!!) Adversarial Robustness for Free!, ICLR, 2023.
> > > >
> > > > [5] DensePure: Understanding Diffusion Models for Adversarial Robustness, ICLR, 2023.
> > > >
> > > > [6] Analysing Diffusion-based Generative Approaches versus Discriminative Approaches for Speech Restoration, ICASSP, 2023.

---

> > > > ### Comment · Reviewer_boQC · 2023-08-19
> > > > **Some comments**
> > > >
> > > > Thank you for your detailed feedback and clarifications. Here are my observations and questions on the topics you highlighted:
> > > >
> > > > **The case of a low SNR**. From my understanding, your adaptive prior essentially alters the sampling trajectory in the second step. The combination of the denoising result from the first step with the weighted average of y forms your adaptive prior. This means that when y's SNR is high, it's logical to integrate y directly into the model input. But when the SNR is low, you still opt for direct integration, with 0.5 being the comprehensive consideration. While this might provide stability, I think the model's performance would undoubtedly be impacted in cases of low SNR. In such situations, wouldn't Eq. 12 be potentially counterproductive?
> > > >
> > > > **Comparison with deterministic models**. Isn't the ultimate goal of designing network structures and solving conditional collapse problems to improve quality? If other works emphasize quality, shouldn't your approach be benchmarked against them for a comprehensive comparison? Also, considering only the single structure of Wavenet is not enough, in my opinion, to fully demonstrate the robustness of your approach, and may lead to different conclusions in other structures.
> > > >
> > > > Your work has certainly piqued my interest, and these observations are meant to better understand the nuances of your methodology and its implications.
> > > >
> > > > Finally, your response did address some of my concerns, and I raised my score to 5.

---

> > > > > ### Author Response · Authors · 2023-08-20
> > > > > **Thanks!!**
> > > > >
> > > > > We appreciate your decision to increase your score!!! Below we answer your new questions:
> > > > > ***
> > > > > > Q1: The case of a low SNR.
> > > > >
> > > > > Yes, you are right. Employing an equal weight provides stability, yet the performance during instances of low SNR would be compromised (albeit still superior to a direct utilization of $y$ as a condition factor, unless the condition optimizer falters). One prospective solution involves introducing an additional adaptive strategy, i.e., $c = \alpha f_{\theta}(y) + (1 - \alpha) y$. We can design an adaptive alpha predictor and hope it can output $\alpha$ based on the quality of raw condition $y$ and samples from condition optimizer. For example, when the condition optimizer produces lower-quality samples, giving more weight to the original condition factor would make sense. Conversely, if the raw condition factor has a low SNR, emphasizing the generated counterpart could be more effective. However, implementing this idea practically is intricate.
> > > > >
> > > > > Given our strong reliance on diffusion-enhanced models to enhance generalization, any new adaptive strategy must be generalizable. For instance, training an adaptive alpha predictor on the (seen) VB-dataset (high SNR \& consistent condition optimizer performance), could lead the model to consistently output higher $\alpha$ values for fusion. Unfortunately, this auxiliary model might not effectively adapt to variations when evaluating the mismatched (unseen) CHIME-4 dataset (low SNR \& potential condition optimizer challenges). To this end, we might need other techniques such as data augmentation and adversarial training to improve its robustness and generalizability. This creates a dilemma: harnessing the speech diffusion model for overarching speech noise reduction generalization while simultaneously necessitating a pre-established generalized model to facilitate its implementation.
> > > > >
> > > > > That's a really good question, and we're working on solving this dilemma. So far, despite our efforts to train a strong alpha predictor, progress has been limited (the alpha predictor is still not generalizable, and the new system has no significant performance improvement over DOSE).
> > > > > ***
> > > > > > Q2: Comparison with deterministic models.
> > > > >
> > > > > Thank you for your valuable comment. When designing our experiments, we aimed to select an existing and widely recognized diffusion enhancement model architecture rather than building a new one, and apply it to all baselines. This approach minimizes alterations to the baselines' original settings and ensures a fair comparison between algorithms unaffected by variations in model architectures. As of now, DiffWave (WaveNet) stands out as the most frequently used model in the realm of diffusion enhancement, and its similarity to WaveNet has shown promising performance in speech enhancement tasks. Apart from the widely used DiffWave, DR-DiffuSE employed a self-constructed model, SGMSE utilized DCUNet, and SGMSE+ employed an existing image diffusion model NCSN++. Our primary concern was that any alteration to the model would necessitate re-tuning all algorithms—a costly endeavor for diffusion enhancement models due to the extensive adjustments required for diffusion parameters. Taking everything into account, we found the use of DiffWave to be the most reasonable choice. This decision rests on both its widely validated effectiveness and the minimal changes it required in the baseline experimental setup.
> > > > >
> > > > > Please note that our intention in sharing the above isn't to contradict your viewpoint; rather, we consider your comment meaningful. We merely wish to explain why our paper exclusively employed DiffWave as the foundational architecture in this version (also to explain DiffWave is not a cherry-picked model). In the revised version, we will consider adding experimental results involving other architectures.
> > > > > ***
> > > > > Thank you once again for your insightful comment! We really enjoy discussing with you!

---

> > > > > > ### Comment · Reviewer_boQC · 2023-08-21
> > > > > > **Some comments on the author's responses**
> > > > > >
> > > > > > **Comment 1**:
> > > > > > I appreciate the authors' detailed discussion regarding the challenges posed by low SNR data. It's evident that the current diffusion models still have room for improvement in this area. The potential solutions proposed by the authors are indeed sensible. I hope the authors can delve deeper into this issue in the supplementary materials and provide a roadmap for future research. Doing so will undoubtedly enhance the impact of this paper.
> > > > > >
> > > > > > **Comment 2**:
> > > > > > My concerns about comparisons with other deterministic models primarily stem from my interest in understanding the generalization capability of the methods presented in the paper. Now that the authors have elaborated on the comparison with other models, I look forward to seeing this discussed in the main text of the paper.
> > > > > >
> > > > > > In conclusion, I would like to express my gratitude to the authors for their comprehensive responses, which have addressed some of my concerns. I believe this paper offers valuable insights, and as a result, I have decided to revise my score from 5 to 6.

---

### Official Review · Reviewer_CtVJ · 2023-07-07

**Soundness:** 3 good
**Presentation:** 3 good
**Contribution:** 3 good
**Rating:** 6
**Confidence:** 3

**Summary:**

This paper proposes a novel model-agnostic approach called DOSE for speech enhancement (SE) using denoising diffusion probabilistic models (DDPMs). In this paper, the authors focus on addressing the challenge of incorporating condition information into DDPMs with two efficient condition-augmentation techniques. Based on the experimental results, the authors claim that the proposed method obtain significant improvements in high-quality and stable speech generation, consistency with the condition factor, and efficiency.

**Strengths:**

1. The proposed method shows good generalization ability with good performance in both matched and mismatched scenarios.

2. This paper shows detailed experimental results and a comprehensive comparison with existing diffusion enhancement methods and deterministic mapping-based method enhancement methods.

3. This paper provides a proper introduction to the problem of condition collapse in generative speech enhancement.

4. This paper is well-written and the flow of the writing is natural so it was easy to read and follow.


**Weaknesses:**

1. To replicate the experiments, more training details and configuration should be provided.

2. It would be better if there were some qualitative analysis in the experiment section.


**Questions:**

Can authors conduct the ablation study to present and analyze the effectiveness of DOSE.

**Limitations:**

The authors do not analyze the limitation of this paper.

---

> ### Author Rebuttal · Authors · 2023-08-04
>
> Thank you for the detailed review and thoughtful feedback. Below we address specific questions.
> ***
> >Q1: To replicate the experiments, more training details and configuration should be provided.
>
> We reported our configurations in Sec 5, line 278-283. We added more experimental details including speech processing, basic architecture, and baseline description in Appendix A.12. As Reviewer k89b suggested, we will update the dataset details in the revision. Note that we also provided the code of DOSE in both supplementary material and anonymous GitHub (Appendix A.1, line 451-452) for replication.
> ***
> >Q2: It would be better if there were some qualitative analysis in the experiment section.
>
> Fully appreciating the question, we'd like to note that there are multiple qualitative results discussed in the Appendix (which we'd be happy to refer to more explicitly in the revision). Specifically:
>
> * We showed visual cases of excessive suppression in A.8;
> * We presented visual cases of error accumulation in A.9;
> * We conducted a counterfactual verification to understand the intrinsic mechanism of DOSE in A.10.
>
> ***
> >Q3: Can authors conduct the ablation study to present and analyze the effectiveness of DOSE.
>
> Thanks for the suggestion! We have conducted ablation studies to quantitatively show the significance of adaptive prior and dropout operation. The results are shown in General Response. We can observe that both of them are crucial for generating consistent samples. We also investigated the significance of adaptive prior and dropout (from other perspectives than metric scores) in Appendix A.6, A.7, and A.10.
> ***
> >Q4: The authors do not analyze the limitation of this paper.
>
> As Reviewer k89b pointed out, we had discussions about limitations in Sec 6 (line 325-338). We also discussed the border impacts in Appendix A.13 (line 675-697). We'd be happy to dedicate a (sub)section for discussing the limitations and the societal impact in the main text.

---

> > ### Comment · Reviewer_CtVJ · 2023-08-20
> >
> > I'd like to thank the authors for their response.

---

### Official Review · Reviewer_Zy9S · 2023-07-10

**Soundness:** 3 good
**Presentation:** 3 good
**Contribution:** 2 fair
**Rating:** 5
**Confidence:** 4

**Summary:**

The authors propose a model-agnostic method called DOSE that employs two efficient condition-augmentation techniques to incorporate condition information into DDPMs for SE. Experiments demonstrate that the approach yields substantial improvements in high-quality and stable speech generation.

**Strengths:**

1. In-depth presentation on diffusion SE, including formulation and methodology.
2. Good results. The authors compare different SE baselines and demonstrate the SOTA results.

**Weaknesses:**

1. It seems that the authors adopt the adaptive prior. Does it only use in the inference process? What is the difference from the adaptive prior in PriorGrad?
2. Why use similarity MOS to evaluate the enhancement model, and what's the difference from MOS?
3. How do you choose p for diffusion dropout? It lacks evaluation and ablation studies on p, which is an important parameter for the proposed diffusion dropout operation.

**Questions:**

If there is a training-inference mismatch? You randomly drop x_t in training, I wonder if it causes the mismatch as the dropout usually does.

**Limitations:**

/

---

> ### Author Rebuttal · Authors · 2023-08-07
>
> Thank you for the detailed review and thoughtful feedback. Below we address specific questions.
>  ***
> > Q1: It seems that the authors adopt the adaptive prior. Does it only use in the inference process? What is the difference from the adaptive prior in PriorGrad?
>
> Yes, the adaptive prior is exclusively used in the inference process. Our method differs from PriorGrad [1] in three key aspects.
>
> * PriorGrad injects instance-level prior knowledge at the initial timestep $T$ and requires modifications to the training process (line 3-5 of Algorithm 1 in [1]). In contrast, our adaptive prior is independent of the training process, allowing our approach to be directly applied to arbitrary pre-trained diffusion enhancement models.
>
> * PriorGrad is designed to speed up the training convergence. Our adaptive prior is used to provide condition knowledge (it can also accelerate inference speed). PriorGrad requires a complete inference process, whereas our approach starts from the intermediate timestep, shortening the sampling trajectory, and thereby improving inference efficiency (i.g., generating clean speech in 2 steps).
>
> * PriorGrad is sensitive to the prior selection -- they have tried several sources of conditional information to compute the prior, but only the normalized frame-level energy of the mel-spectrogram worked (cf. Sec 4.1 in [1]). This means that it is hard for developers to choose an appropriate adaptive prior. In contrast, our adaptive prior, computed directly from noisy speech, is stable and effective.
>
> We appreciate your question and hope this explanation addresses your concerns.
> ***
> > Q2: Why use similarity MOS to evaluate the enhancement model, and what's the difference from MOS?
>
> As the primary focus of our work is to tackle the condition collapse problem in diffusion enhancement models, it is crucial to assess the consistency of the generated speech with real speech. While MOS is commonly employed to rate the overall naturalness and fluency of synthesized audio in speech synthesis, we introduce another metric, called similarity MOS, which specifically evaluates the consistency (content, timbre, emotion, and prosody) between the generated speech and the real speech. We provided details about subjective human evaluation in Appendix A.3 (line 486-496).
> ***
> > Q3: How do you choose $p$ for diffusion dropout? It lacks evaluation and ablation studies on $p$, which is an important parameter for the proposed diffusion dropout operation.
>
> Similar to the process of selecting $\tau_1$ and $\tau_2$, we determine the optimal values for $p$ by evaluating the performance on a validation dataset. In this study, we pre-defined a relatively coarse candidate set $\{0, 0.1, 0.5, 0.9\}$ for $p$ and found that $p=0.5$ generated appealing results. We have included supplementary ablation studies for the hyper-parameter $p$ in the above table (in General Response). Note that we presented parameter sensitivity experiments in Appendix A.7, and the influence of $p$ (line 589-604) was shown in Figure 11 and Figure 12. We will mention this explicitly in the main text.
> ***
> > Q4: If there is a training-inference mismatch? You randomly drop $x_t$ in training, I wonder if it causes the mismatch as the dropout usually does.
>
> Great question! It is essential to clarify that our method is different from the conventional dropout technique typically applied to neural networks. In conventional dropout, random units (neurons) or their activations are dropped out during training to prevent overfitting and improve generalization. In contrast, our approach focuses on randomly dropping out $x_t$ (input features) to mitigate the condition collapse problem in diffusion enhancement models. This strategy would force the model to generate condition-consistent samples. While dropout might introduce a slight training-inference mismatch, we carefully validate our model's performance on benchmark datasets and ensure that it does not significantly affect the quality of the generated samples. We will include a more detailed explanation of the dropout technique and its implications in our revision.

---

### Author Rebuttal · Authors · 2023-08-07

We would like to thank all reviewers for providing high-quality reviews and insightful feedback.

---

We are encouraged that reviewers think our paper "provides valuable insights into the functioning of the denoising diffusion probabilistic model for speech enhancement'' (R4), "an interesting and promising solution to condition collapse issue'' (R3, R4), "technically/theoretically in-depth'' (R1, R5), "comprehensive experiments and thorough analysis'' (R2, R3, R4, R5), "broader implications and versatility" (R4), and "well-written'' (R2, R4).

(We abbreviate the reviewer Zy9S, CtVJ, boQC, k89b, RTAK to R1, R2, R3, R4, R5, respectively.)

---

We provide additional ablation results requested by R1, R2 and R3, shown in the table below -- $p$ denotes the dropout rate, $\epsilon$ and $x$ denote different training objectives, and the number of steps denotes how many steps are needed to generate speech during the inference stage.

|  |  |  | VoiceBank |  |  |  |  | CHIME-4 |  |  |
|---|:---:|:---:|:---:|:---:|:---:|:---:|:---:|:---:|:---:|:---:|
| Variable | STOI(%) | PESQ | CSIG | CBAK | COVL | STOI(%) | PESQ | CSIG   | CBAK | COVL |
| p = 0 ( $\epsilon$ ) 2 steps (to R2) | 92.7    | 2.13 | 3.44 | 2.58 | 2.76 | 86.5    | 1.39 | 2.69   | 2.04 | 1.98 |
| p = 0 ( $x$ ) 2 steps (to R1) | 93.3    | 2.49 | 3.74 | 3.03 | 3.10 | 80.6    | 1.37 | 2.59   | 2.06 | 1.92 |
| p = 0.1 ( $x$ ) 2 steps (to R1) | 93.5 | 2.50 | 3.66 | 3.24 | 3.08 | 82.0    | 1.44 | 2.69   | 2.10 | 2.00 |
| p = 0.5 ( $x$ ) 2 steps (to R1) | **93.6** | **2.56** | **3.83** | **3.27** | **3.19** | **86.6** | **1.52** | **2.71** | **2.15** | **2.06** |
| p = 0.9 ( $x$ ) 2 steps (to R1) | 92.6 | 2.33 | 3.54 | 3.01 | 2.93 | 83.3 | 1.36 | 2.67 | 2.02 | 1.95 |
| p = 0.5 ( $x$ ) 6 steps (to R3) | 93.1 | **2.56** | 3.78 | 3.03 | 3.16 | 82.2 | 1.43 | 2.70 | 2.12| 2.01 |
| p = 0.5 ( $x$ ) 50 steps (to R2 and R3) | 93.2 | 2.48 | 3.66 | 3.18 | 3.06 | 82.1 | 1.39 | 2.49 | 2.01 | 1.87 |

---

We list all needed references here to facilitate the subsequent point-to-point rebuttals.

[1] PriorGrad: Improving Conditional Denoising Diffusion Models with Data-Dependent Adaptive Prior, ICLR 2021.

[2] SDEdit: Guided Image Synthesis and Editing with Stochastic Differential Equations, ICLR, 2021.

[3] DifFace: Blind Face Restoration with Diffused Error Contraction, Arxiv, 2022.

[4] Diffsinger: Singing voice synthesis via shallow diffusion mechanism, AAAI, 2022.

[5] A Study on Speech Enhancement Based on Diffusion Probabilistic Model, APSIPA, 2021.

[6] (Certified!!) Adversarial Robustness for Free!, ICLR, 2023.

[7] DensePure: Understanding Diffusion Models for Adversarial Robustness, ICLR, 2023.

[8] DiffWave: A Versatile Diffusion Model for Audio Synthesis, ICLR, 2021.

[9] Wavenet: A Generative Model for Raw Audio, Arxiv, 2016.

[10] A Wavenet for Speech Denoising, ICASSP, 2018.

[11] End-to-End Music Source Separation: Is It Possible in the Waveform Domain?, Interspeech, 2019.

[12] Conditional Diffusion Probabilistic Model for Speech Enhancement, ICASSP, 2022.

[13] Analysing Diffusion-based Generative Approaches Versus Discriminative Approaches for Speech Restoration, ICASSP, 2023.

[14] Score-Based Generative Modeling through Stochastic Differential Equations, ICLR, 2021.

[15] Generative Modeling by Estimating Gradients of the Data Distribution, NeurIPS, 2019.

[16] A Recursive Network with Dynamic Attention for Monaural Speech Enhancement, Interspeech, 2020.

[17] A Time-domain Monaural Speech Enhancement with Feedback Learning, APSIPA, 2020.

[18] Densely Connected Progressive Learning for LSTM-based Speech Enhancement, ICASSP, 2018.

[19] A Multi-Target SNR-Progressive Learning Approach to Regression Based Speech Enhancement, TASLP, 2020.

 [20] DiffRoll: Diffusion-based Generative Music Transcription with Unsupervised Pretraining Capability, ICASSP, 2023.

---

### Author Response · Authors · 2023-08-17
**Polite Reminder about Author-Reviewer Discussion Deadline (Aug 21st)**

Hi everyone,

Thank you again for the thoughtful reviews. We wanted to kindly remind you that the author-reviewer discussion deadline is approaching (Aug 21st). We were hoping to engage in some discussion regarding your comments, to make sure we have enough time to provide our input where it is needed. Thanks again & looking forward to discussing!

Best, Authors

---

### Decision · Program_Chairs · 2023-09-21

**Decision:**

Accept (poster)

**Comment:**

All reviewers see the merit of the paper and recommend acceptance. The approach is simple and intuitive, and the experiments are sound. The writing, however, is overly complicated.  Some revision to improve the readability and accessibility of the paper, such as front-loading the topic sentence in each paragraph, is recommended.